# FOSL: A Foldable Sparse-and-Low-Rank Method for Efficient LLM Pre-training

## Abstract

We propose FOSL, a foldable, sparse-and-low-rank reparameterization for efficient pre-training that decouples compute from width. Each linear / FFN / attention projection is rewritten as two cooperating paths: a low-rank path that injects expressive features via a compact adapter, and a folded sparse path that computes only a subset of output channels and synthesizes the remainder as virtual channels by reusing computed ones. A lightweight, variance-preserving rescaling keeps activations stable when channels are reused multiple times. This design delivers the benefits of a narrower network internally while maintaining full-width activations at the interface, avoiding the representation bottlenecks of hard pruning and complementing low-rank-only approaches. We evaluate FOSL for LLM pre-training across model scales from 60M to 7B parameters. Our experiments demonstrate that FOSL matches or surpasses full-rank models, while substantially reducing memory and computation costs.

## 1 Introduction

Large language models (LLMs) have rapidly advanced capabilities across language, speech, and vision. Yet the prevailing recipe—ever wider layers trained on ever larger corpora—pushes pre-training compute, memory footprint, and energy consumption to unsustainable levels (Bhardwaj et al., 2025). Thus, beyond deployment-time compression, there is a pressing need for training-time parameterizations that preserve model expressivity at a fraction of the cost.

Parameter-efficient tuning in the fine-tuning regime popularized by Low-Rank Adaptation (LoRA) freezes pre-trained weights and learns low-rank adapters, dramatically reducing trainable parameters without altering the backbone weights (Hu et al., 2021). Extending low-rank approach to pre-training is less straightforward: naïvely constraining updates or weights to a fixed low-rank subspace often harms optimization and final quality (Wang et al., 2025b; Kamalakara et al., 2022).

This area has recently attracted extensive research on efficient pre-training, with three complementary directions emerging. The first group of approaches extend the idea of low-rank LoRA-style updates. ReLoRA periodically merge accumulated low-rank updates to recover an effectively high-rank model while continuing to train through low-rank adapters, enabling high-rank training dynamics with lower optimizer memory (Lialin et al., 2023). SwitchLoRA increases the effective update rank by frequently and smoothly switching LoRA subspaces, targeting only a few dimensions at a time to maintain optimizer-state consistency (Zhou et al., 2025). LORO directly parameterizes each weight as $W = BA$ and optimizes on the rank $r$ manifold via a Low-rank Riemannian Optimizer (Mo et al., 2025). By jointly updating the factor pair $(B, A)$ so that their product follows the Riemannian steepest-descent direction, without ever forming full-size matrices or gradients.

Instead of constraining weights, another line of work reduces optimizer memory by projecting gradients (Khodak et al., 2021; Chen et al., 2024; Zhang et al., 2024). GaLore projects gradients onto a low-rank subspace to cut optimizer state memory while still updating all parameters (Zhao et al., 2024). Building on this, Fira argues that one can preserve a low-rank optimizer while approximating full-rank training via norm-based scaling and a norm-growth limiter to suppress loss spikes, often matching or surpassing full-rank baselines (Chen et al., 2024).

A third direction modifies network structure itself, often combining low-rank reparameterization with sparsity. CoLA factorizes linear blocks and inserts nonlinearity between factors, enforcing

low-rank structure in activations (Liu et al., 2025); it reports simultaneous FLOP and parameter reductions with full-rank-level perplexity. In parallel, SLTrain parameterizes weights as low-rank + (fixed-support) sparse components and shows substantial memory savings with quality competitive to full-rank pre-training (Han et al., 2024; Glentis et al., 2025). LOST co-designs the low-rank and structured sparse components: it initializes them via SVD so that the sparse residual complements the dominant low-rank subspace, improving trainability under tight efficiency budgets Li et al. (2025). More related work is covered in Appendix G.

Despite this progress, a central tension remains: *How can we reduce true compute (FLOPs / activation memory / communication) while keeping the forward signal space rich enough to emulate a wide, full-rank model?* Low-rank parameterizations shrink the search space; sparsity prunes degrees of freedom. Both can under-represent the effective rank of intermediate signals unless carefully compensated. Intriguingly, post-training model folding (Wang et al., 2025a) shows that many channels/heads in pre-trained networks are redundant up to similarity: simply clustering channels (FFN) or heads (self-attention) and averaging within clusters, without data or fine-tuning, can yield competitive compressed models when paired with variance-preserving corrections (Jordan et al., 2022). This suggests a different approach to achieve training-time efficiency: *exploit channel-wise redundancy not by deleting channels, but by explicitly reusing them* during training.

We introduce FOSL—a foldable, sparse-and-low-rank pre-training scheme, which enables training LLMs with low-rank weights while maintaining performance as good as full-rank training, as shown in Fig. 1. At a high level, FOSL reparameterizes each linear/FFN/attention projection with two cooperating paths: (i) *a low-rank path* that injects expressive features through a compact adapter, and (ii) *a folded sparse path* that computes only a subset of output channels and synthesizes the remainder as virtual channels by reusing the computed outputs. A lightweight variance-stabilization keeps activations well behaved when a channel is reused multiple times. The result is a model that preserves full-width activations (and compatibility with dense kernels) while paying the compute of a much narrower layer. Unlike hard pruning, FOSL retains a rich forward signal. Unlike low-rank-only updates, it amortizes width via learned reuse. Our contributions are:

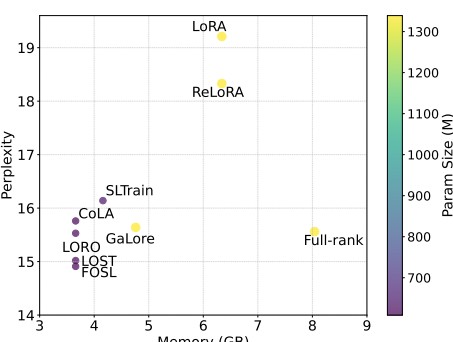

Figure 1: Performance comparison of pre-training methods on LLaMA-1B (C4 dataset). Smaller circles in the lower-left indicate better memory efficiency and lower perplexity. See Tab. 2 for the complete list of results.

- We propose a training-time parameterization through foldable structured sparsity (FOSL) that computes only a subset of output channels and folds the rest by reusable copies.

- We propose a novel variance correction method that preserves the effective width of the original layer, enabling stable statistics under reuse while maintaining the compute and memory efficiency of narrow layers.

- We demonstrate that FOSL surpasses the performance of SoTA efficient LLM pre-training with fewer number of parameters, particularly in pre-training LLaMA 60m to 1B. We also show competitive or superior results of FOSL on efficient fine-tuning tasks.

## 2  LOW-RANK AND SPARSE PRE-TRAINING

**Preliminaries.**  Let $x \in \mathbb{R}^d$ be the input to a linear map and $W \in \mathbb{R}^{m \times d}$ the weight of a projection in an FFN or in an attention block (*e.g.*, $W_Q, W_K, W_V, W_O$). The output is $y = Wx$. Transformer pre-training minimizes an autoregressive loss $\mathcal{L}(\Theta; \mathcal{D})$ over parameters $\Theta$ on a large corpus $\mathcal{D}$. We focus on parameterizations of $W$ that reduce *true compute* (both FLOPs and activation memory) and optimizer-state memory while preserving expressivity.

## 2.1 Low-rank parameterizations for pre-training

A standard approach constrains $W$ to the product of two thin factors:

$$W = BA^\top, \qquad B \in \mathbb{R}^{m \times r}, \ A \in \mathbb{R}^{d \times r}, \ r \ll \min\{m, d\}. \tag{1}$$

The forward becomes $y = B(A^\top x)$, cutting multiplies from $md$ to $r(m+d)$ and reducing optimizer states to the size of $(A, B)$. While this can substantially save memory and compute, fixing a global rank $r$ limits the search space and often underperforms full-rank pre-training.

**LoRA-style update parameterization.** LoRA (Hu et al., 2021) popularized *update-level* low rank for fine-tuning by freezing a baseline $W_0$ and learning a low-rank update $\Delta W = BA^\top$:

$$W = W_0 + \Delta W, \qquad \text{rank}(\Delta W) \leq r. \tag{2}$$

When adapted to pre-training from scratch, one may (i) initialize $W_0$ and only train the low-rank branch (periodically merging) (Lialin et al., 2023; Zhou et al., 2025), or (ii) drop $W_0$ and train the factorized $BA^\top$ directly (as in Eq. 1) (Hu et al., 2021; Mo et al., 2025). These variants reduce optimizer memory but may still exhibit a representation bottleneck for wide LLMs unless complemented by additional mechanisms (*e.g.*, periodic merges or complementary components).

## 2.2 Sparse component in low-rank pre-training

Both *SLTrain* (Han et al., 2024) and *LOST* (Li et al., 2025) enrich low-rank models by adding a sparse component to recover information outside the dominant low-rank subspace. The generic form is

$$W = BA^\top + S, \qquad \text{rank}(BA^\top) \leq r, \quad S \text{ is sparse}, \tag{3}$$

trading a small number of extra parameters and cheap sparse operations for improved expressivity.

**SLTrain (unstructured, fixed-support).** SLTrain samples a *fixed* unstructured support $\Omega \subseteq [m] \times [d]$ once and only trains $S_{ij}$ on $\Omega$ (Han et al., 2024):

$$\text{supp}(S) = \Omega, \ |\Omega| = k, \qquad \min_{A,B,S_\Omega} \mathcal{L}\big((BA^\top + S)x; \mathcal{D}\big). \tag{4}$$

This avoids index churn and keeps optimizer states small. With LoRA-style initialization (*e.g.*, Kaiming for $A$, zeros for $B$), the combination captures dominant modes via $BA^\top$ and adds entrywise corrections via $S$, typically yielding performance closer to full-rank at memory comparable to pure low-rank (params $r(m+d) + k \ll md$). Forward cost is two thin GEMMs plus a sparse add for $Sx$.

**LOST (structured, SVD-guided co-design).** LOST co-designs the two parts so that they occupy *complementary* subspaces. Starting from the initialized $W^{(0)}$, it performs a one-shot SVD $W^{(0)} = U\Sigma V^\top$, forms the initial low-rank $W^{(0)}_{\text{low}} = \sum_{i=1}^r \sigma_i u_i v_i^\top$, and derives a *structured* (channel-wise) sparse component from the residual spectrum (Li et al., 2025):

$$W = W_{\text{low}} + \gamma W_{\text{sp}}, \qquad \text{supp}(W_{\text{sp}}) \in \mathcal{C}_{\text{channel}}, \tag{5}$$

where $\gamma$ balances the two and $\mathcal{C}_{\text{channel}}$ denotes the allowed nonzero pattern (support set), *e.g.*, per-output channel, and is hardware-friendly. By steering $W_{\text{sp}}$ toward the complement of the top-$r$ subspace, LOST preserves trainability and effective signal rank under tight efficiency budgets. Some variants optionally insert a nonlinearity between factors $x \mapsto B\phi(A^\top x)$ to enhance functional expressivity without abandoning low-rank compute.

Both methods have limitations: SLTrain fixes an unstructured support that cannot adapt and is hardware-unfriendly, while LOST relies on an offline SVD initialization, enforces less flexible channel-wise sparsity, and only initializes (but does not maintain) subspace complementarity during training.

## 3 Foldable and Sparse Low-Rank Pre-Training

FOSL is inspired by model folding (Wang et al., 2025a) – a recently introduced post–training, data-and fine-tuning–free compression technique that exploits channel/head redundancy by clustering

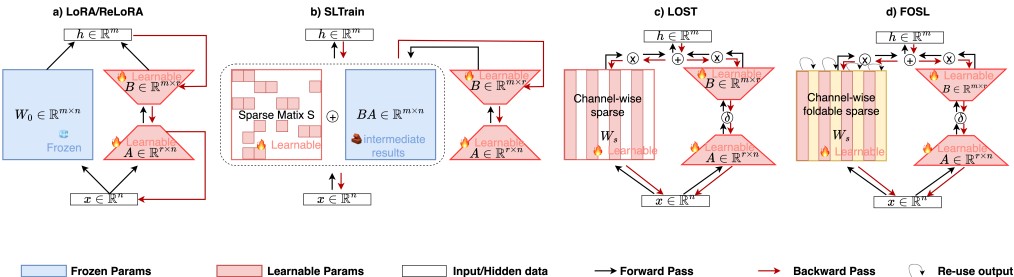

Figure 2: Comparison between different efficient pre-training frameworks. a) LoRA / ReLoRA (Lialin et al., 2023) freezes a full-rank weight; b) SLTrain (Han et al., 2024) requires reconstruction of the low-rank and sparse matrices; c) LOST (Li et al., 2025) utilizes structurally sparse $W$; d) Our FOSL introduces foldable structured sparsity.

weight vectors within a layer and merging members of each cluster into a representative (*e.g.*, the cluster centroid). Proceeding layer-by-layer, folding shrinks the effective width and FLOPs while keeping tensor interfaces compatible with the dense model, and has been shown to retain accuracy on CNNs and LLMs (*e.g.*, ResNet18 (He et al., 2015b), LLaMA-7B (Touvron et al., 2023b;a)). The observation that many channels are redundant up to similarity motivates our foldable structured sparsity method with principled variance stabilization: instead of deleting channels during training, we reuse them to maintain full-width activations at the interface while paying the compute of a narrower branch.

**Design goal.** Given an $m \times d$ projection with output width $m$, we seek a parameterization that keeps the *interface activations* full-width while paying the compute of a much narrower layer. To this end, we decompose the layer output into a *low-rank path* and a *folded sparse path*:

$$y = \alpha\, y_{\text{lr}}(x) + \beta\, y_{\text{fold}}(x) + b, \qquad \alpha, \beta \in \mathbb{R}, \tag{6}$$

where $b \in \mathbb{R}^m$ and the two paths are computed with low true compute and small optimizer footprint. We subsequently reparameterize this mixing as $y = \gamma\, y_{\text{lr}}(x) + (1 - \gamma)\, y_{\text{fold}}(x) + b$ with $\gamma \in [0, 1]$ (either fixed or trainable, see Tab. 4). We consider three mixing methods: (i) the fixed mixing coefficients $(\alpha, \beta) = (0.7, 0.3)$, following (Li et al., 2025); (ii) a trainable per-layer scalar $\gamma$ with a bounded parameterization $\gamma = \sigma(\theta)$; and (iii) a trainable per-channel vector $\gamma \in [0, 1]^m$ applied at the addition. Detailed evaluation is shown in Tab. 4.

### 3.1 LOW-RANK PATH

Consistent with Sec. 2.1, we form a compact adapter with $A \in \mathbb{R}^{d \times r}$ and $B \in \mathbb{R}^{m \times r}$, $r \ll \min\{m, d\}$. FOSL uses a lightweight nonlinearity between the factors:

$$y_{\text{lr}}(x) = B\, \phi(A^\top x) \cdot s, \tag{7}$$

where $\phi$ is an element-wise activation (SiLU by default, following the spirit of CoLA (Liu et al., 2025)) and $s$ is a scalar (fixed $=$ `lora_alpha`$/r$ or trainable with a bounded parameterization). Eq. 7 preserves the compute profile of rank-$r$ adapters (two thin GEMMs) while enriching the function class beyond a purely linear rank-$r$ map.

### 3.2 FOLDED SPARSE PATH

Let the *folding ratio* be $\rho \in (0, 1]$ and define

$$m_{\text{fold}} = \lfloor \rho m \rfloor, \qquad m_{\text{base}} = m - m_{\text{fold}}.$$

The folded path computes only $m_{\text{base}}$ *real* channels and synthesizes the remaining $m_{\text{fold}}$ as *virtual* copies:

$$z = W_{\text{base}}\, x, \qquad W_{\text{base}} \in \mathbb{R}^{m_{\text{base}} \times d}, \tag{8}$$

$$y_{\text{fold,raw}} = D_\pi\, z, \qquad D_\pi \in \{0, 1\}^{m \times m_{\text{base}}}. \tag{9}$$

Here $D_\pi$ is a *duplication* (reuse) matrix determined by a reuse map $\pi : \{1, \ldots, m\} \to \{1, \ldots, m_{\text{base}}\}$. For each output position $j \in \{1, \ldots, m\}$, the $j$-th row of $D_\pi$ is the one-hot vector $e_{\pi(j)}^\top$. For the $m_{\text{base}}$ real positions we set $\pi(j) = j$ (*i.e.*, identity), while for the $m_{\text{fold}}$ virtual positions we *reuse* selected real channels (possibly with repetitions). This yields full-width outputs without computing $m_{\text{fold}}$ additional dot-products.

**Initialize the reuse map.** The reuse map $\pi$ is sampled uniformly given $m_{\text{base}}$ and $m_{\text{fold}}$. Let $n = m_{\text{fold}}$ be the number of virtual outputs. We draw $k = \lceil n/m_{\text{base}} \rceil$ independent random permutations of $\{1, \ldots, m_{\text{base}}\}$, concatenate them, and assign the first $n$ indices to the $n$ virtual positions. This "multi-permutation concatenation" keeps reuse balanced: each real channel $i$ is selected either $\lfloor n/m_{\text{base}} \rfloor$ or $\lceil n/m_{\text{base}} \rceil$ times. The reuse count is $k_i = |\{j > m_{\text{base}} : \pi(j) = i\}|$, so channel $i$ appears in $1+k_i$ positions including its own. We fix the reuse map $\pi$ during training; end-to-end learning of $\pi$ (*e.g.*, via differentiable assignment or periodic re-matching) is left to future work.

**Variance correction under reuse.** If a real channel $i$ is reused $k_i$ times (so it appears in $1+k_i$ positions counting itself), naively duplicating $z_i$ would inflate its total second moment by a factor of $1+k_i$. We mitigate this with a diagonal rescaling on the real channels:

$$C = \text{diag}\Big((1+k_1)^{-1/2}, \ldots, (1+k_{m_{\text{base}}})^{-1/2}\Big), \qquad y_{\text{fold}} = D_\pi \, C \, z. \tag{10}$$

Thus each copied position becomes $(D_\pi C z)_j = (1+k_i)^{-1/2} z_i$ for some $i = \pi(j)$, so that individual copies have reduced variance while the aggregate energy of each real channel is preserved.

**Lemma 1** (Energy-preserving scaling under reuse). *Assume each real channel $z_i$ is zero-mean with identical variance $\sigma^2$. Suppose channel $i$ is reused $k_i$ times (so it appears in $1+k_i$ output positions), and define $C_{ii} = (1+k_i)^{-1/2}$ as in Eq. 10. Then every copy associated with channel $i$ inside $y_{\text{fold}} = D_\pi C z$ satisfies*

$$\text{Var}\big((D_\pi C z)_j\big) = \tfrac{1}{1+k_i} \sigma^2 \quad \text{for all } j \text{ with } \pi(j) = i,$$

*and the aggregate variance across the $1+k_i$ positions equals that of the original channel:*

$$\sum_{j: \, \pi(j)=i} \text{Var}\big((D_\pi C z)_j\big) = \sigma^2.$$

*Proof.* Each copied position equals $(D_\pi C z)_j = (1+k_i)^{-1/2} z_i$ for some $i = \pi(j)$. Hence $\text{Var}\big((D_\pi C z)_j\big) = (1+k_i)^{-1}\text{Var}(z_i) = (1+k_i)^{-1}\sigma^2$, and summing over the $1+k_i$ copies yields $\sum_{j: \, \pi(j)=i} \text{Var}\big((D_\pi C z)_j\big) = \sigma^2$. $\square$

*Remark.* In our implementation, Eq. 10 is applied to the pre-activations $z = W_{\text{base}} x$ after LayerNorm. Under the common mean-field approximation that post-LayerNorm channels have comparable variance and weak cross-channel correlations, this energy-preserving scaling keeps the overall scale of $y_{\text{fold}}$ comparable to the dense model while preventing the $(1+k_i)$-fold variance blow-up that would arise from naive reuse. Heavy-tailed or strongly dependent activations may require more expressive corrections; our diagonal scheme should therefore be viewed as a practically effective second-moment approximation rather than an exact distributional match.

**Backpropagation through folding.** Let $\bar{y}_{\text{fold}} = \partial\mathcal{L}/\partial y_{\text{fold}}$. From Eq. 10, the gradient w.r.t. real outputs $z$ accumulates from all their copies:

$$\frac{\partial\mathcal{L}}{\partial z} = C \, D_\pi^\top \, \bar{y}_{\text{fold}}, \tag{11}$$

so each $z_i$ receives the (scaled) sum of gradients from its virtual replicas. The backward cost remains that of a *narrow* projection (size $m_{\text{base}}$), plus negligible gather/scatter.

Table 1: Per-layer complexity for a projection $W \in \mathbb{R}^{m \times d}$. Here $r \ll \min\{m, d\}$; for SLTrain/LOST, $\zeta = \mathrm{nnz}(S)/(md)$ is the (learned) sparse ratio. For FOSL, $\rho$ is the folding ratio of *virtual* outputs so that $m_{\text{base}} = (1 - \rho)m$. "Optimizer state" scales like parameters (e.g., $\approx 2\times$ for Adam) and we report big-$\mathcal{O}$ counts. All FLOP counts are reported *per output vector*; for a batch of size $B$ and sequence length $L$, they are multiplied by a common factor $BL$ shared by all methods.

| Method | Trainable Params | Optim. State | Fwd FLOPs | Bwd FLOPs |
|---|---|---|---|---|
| Full-rank (baseline) | $\mathcal{O}(md)$ | $\mathcal{O}(md)$ | $\mathcal{O}(md)$ | $\mathcal{O}(md)$ |
| LoRA / ReLoRA | $\mathcal{O}(r(m+d))$ | $\mathcal{O}(r(m+d))$ | $\mathcal{O}(md + r(m+d))$ | $\mathcal{O}(md + r(m+d))$ |
| SLTrain (lr+unstruct. sparse) | $\mathcal{O}(r(m+d) + \zeta\,md)$ | $\mathcal{O}(r(m+d) + \zeta\,md)$ | $\mathcal{O}(r(m+d) + \zeta\,md)$ | $\mathcal{O}(r(m+d) + \zeta\,md)$ |
| LOST (lr+struct. sparse) | $\mathcal{O}(r(m+d) + \zeta\,md)$ | $\mathcal{O}(r(m+d) + \zeta\,md)$ | $\mathcal{O}(r(m+d) + \zeta\,md)$ | $\mathcal{O}(r(m+d) + \zeta\,md)$ |
| **FOSL** (ours) | $\mathcal{O}(r(m+d) + (1-\rho)\,md)$ | $\mathcal{O}(r(m+d) + (1-\rho)\,md)$ | $\mathcal{O}(r(m+d) + (1-\rho)\,md)$ | $\mathcal{O}(r(m+d) + (1-\rho)\,md)$ |

### 3.3 PUTTING IT TOGETHER AND COMPLEXITY

Combining Eq. 6–Eq. 10, the FOSL layer is

$$y = \alpha\, B\, \phi(A^\top x)\, s + \beta\, D_\pi\, C\, (W_{\text{base}} x) + b, \qquad (12)$$

where the low-rank path uses a compact adapter with a lightweight nonlinearity between the factors (in the spirit of CoLA), and the folded path computes only $m_{\text{base}}$ *real* channels while synthesizing the remaining $m_{\text{fold}}$ outputs by reuse. Let the folding ratio be $\rho \in [0, 1)$ with $m_{\text{fold}} = \lfloor \rho m \rfloor$ and $m_{\text{base}} = m - m_{\text{fold}} = (1 - \rho)m$. The duplication operator $D_\pi$ is an index mapping (not a trained matrix), and $C = \mathrm{diag}\big((1 + k_i)^{-1/2}\big)$ stabilizes variance when channel $i$ is reused $k_i$ times. The interface remains full width ($\mathbb{R}^m$), so dense kernels and downstream tensor shapes are unchanged, while the *true compute* scales with $r$ and $m_{\text{base}}$. See Fig. 2 for a high-level comparison with related efficient pre-training frameworks. Further theoretical support for the folded path, including gradient aggregation, isometry, and SNR gains—is provided in Appendix B.

**Parameters, optimizer state, and FLOPs.** The trainable parameters of FOSL are $A \in \mathbb{R}^{d \times r}$, $B \in \mathbb{R}^{m \times r}$, $W_{\text{base}} \in \mathbb{R}^{m_{\text{base}} \times d}$, plus small scalars/vectors (*e.g.*, LoRA scale*s*, bias $b$). The duplication map $\pi$ and counts $\{k_i\}$ are stored as indices/counters ($\mathcal{O}(m)$) and do not introduce optimizer states. Forward cost is two thin GEMMs for the low-rank path and one narrow GEMM for the folded path; gather/scatter from $D_\pi$ and diagonal $C$ are memory-light. Backward is of the same order as forward (up to constant factors suppressed in big-$\mathcal{O}$). See Tab. 1 for a per-layer big-$\mathcal{O}$ comparison against baselines. For FOSL the asymptotics scale as $\mathcal{O}\big(r(m+d) + (1 - \rho)md\big)$ with $m_{\text{base}} = (1 - \rho)m$.

**Connection to prior sections.** With $\phi \equiv \mathrm{id}$ and $g \equiv 1$, FOSL implements a linear map $W_{\text{eff}} = \alpha\, BA^\top + \beta\, D_\pi C W_{\text{base}}$ whose rank satisfies $\mathrm{rank}(W_{\text{eff}}) \leq r + m_{\text{base}}$. Thus, unlike $W = BA^\top + S$ in Eq. 3 (where $S$ is a freely learned sparse residual), FOSL attains similar *effective* capacity using a narrow compute branch ($m_{\text{base}}$) plus channel reuse—preserving an $m$-dimensional interface while reducing true compute to that of width $r + m_{\text{base}}$.

## 4 EXPERIMENTS

### 4.1 LLMS PRE-TRAINING

**Experimental setup**. We pre-train on the Colossal Clean Crawled Corpus (C4) (Raffel et al., 2020), a large-scale, cleaned and deduplicated snapshot of Common Crawl widely used for language modeling. Following the experimental protocols of *SLTrain* (Han et al., 2024), the benchmarking in (Glentis et al., 2025), and *LOST* (Li et al., 2025), we adopt the same pre-training setup and token budgets per model size. Concretely, we train LLaMA-family models from 60M, 130M, 350M, up to 1B parameters (Touvron et al., 2023b;a), using the same number of tokens as prior work for each size (see Tab. 2). Baselines include *Full-Rank*, *GaLore* (Zhao et al., 2024), *Fira* (Chen et al., 2024), *SLTrain* (Han et al., 2024), *CoLA* (Liu et al., 2025), *LORO* (Mo et al., 2025), and *LOST* (Li et al., 2025). For our method, we reparameterize all linear projections in LLaMA (self-attention $W_Q/W_K/W_V/W_O$ and MLP/FFN blocks) with FOSL, keeping the external width unchanged while reducing true compute; unless otherwise noted, we use the same AdamW optimizer, schedules,

| | 60M | | | 130M | | | 350M | | | 1B | | |
|---|---|---|---|---|---|---|---|---|---|---|---|---|
| $r$ / $d$ | 128 / 512 | | | 256 / 768 | | | 256 / 1024 | | | 512 / 2048 | | |
| Tokens | 1.4B | | | 2.6B | | | 7.8B | | | 13.1B | | |
| | PPL | Param | Mem | PPL | Param | Mem | PPL | Param | Mem | PPL | Param | Mem |
| Full-Rank | 30.27 | 58 | 0.35 | 23.13 | 134 | 0.81 | 18.76 | 368 | 2.21 | 16.52 | 1339 | 8.04 |
| LoRA | 35.30 | 43 | 0.36 | 25.07 | 94 | 0.84 | 19.13 | 185 | 1.85 | 15.83 | 609 | 6.34 |
| Low-Rank | 35.13 | 43 | 0.24 | 26.71 | 94 | 0.57 | 21.77 | 185 | 1.11 | 18.22 | 609 | 3.66 |
| GaLore (Zhao et al., 2024) | 34.58 | 58 | 0.28 | 25.31 | 134 | 0.61 | 19.37 | 368 | 1.59 | 15.57 | 1339 | 4.76 |
| Fira (Chen et al., 2024) | 30.34 | 58 | 0.28 | 22.96 | 134 | 0.61 | 16.82 | 368 | 1.59 | 15.10 | 1339 | 4.76 |
| SLTrain (Han et al., 2024) | 32.58 | 47 | 0.30 | 24.17 | 104 | 0.67 | 18.59 | 215 | 1.54 | 15.40 | 732 | 5.33 |
| LORO Mo et al. (2025) | 33.87 | 43 | 0.24 | 24.78 | 94 | 0.57 | 19.66 | 185 | 1.11 | 15.53 | 609 | 3.66 |
| CoLA Liu et al. (2025) | 34.10 | 43 | 0.24 | 25.61 | 94 | 0.57 | 19.75 | 185 | 1.11 | 15.76 | 609 | 3.66 |
| LOST Li et al. (2025) | 32.25 | 43 | 0.24 | 24.05 | 94 | 0.57 | 18.95 | 185 | 1.11 | 15.02 | 609 | 3.66 |
| **FOSL** | $33.28_{\pm 0.21}$ | 43 | 0.24 | $24.15_{\pm 0.18}$ | 94 | 0.57 | $18.65_{\pm 0.09}$ | 185 | 1.11 | $14.87_{\pm 0.03}$ | 609 | 3.66 |

Table 2: Comparison of validation perplexity, parameter count in millions (M), and estimated memory in gigabytes (GB) across methods. $r$ and $d$ denote the target rank and the model's hidden dimension, respectively. All FOSL models in this table use a trainable layer-wise mixing coefficient and sparsity $\rho = 0.99$, with ranks chosen so that the total parameter count matches other baslines at each scale; no LOST-style SVD initialization is used. FOSL results are averaged over 3 independent runs, with standard deviations indicated. Results for other methods are taken from (Glentis et al., 2025; Li et al., 2025; Mo et al., 2025; Han et al., 2024). SLTrain and LOST also use a sparsity of $\rho = 0.99$.

warming up ratio, and data packing as in these prior works (Zhao et al., 2024; Glentis et al., 2025). More details of implementation can be found in Appendix A.

As summarized in Tab. 2, FOSL delivers favorable perplexity–efficiency trade-offs across LLaMA-60M/130M/350M/1B under matched token budgets and ranks. At 1B, FOSL attains the best validation perplexity (PPL 14.69), outperforming the dense baseline (16.52) and strong efficient pre-training baselines such as Fira (15.10) and LOST (15.02), while using 730M parameters and an estimated 4.08 GB of memory—approximately 45% fewer parameters and 49% less memory than Full-Rank. At 350M, FOSL also surpasses Full-Rank (18.23 vs 18.76 PPL) with roughly 46% lower memory (1.20 vs 2.21 GB). At the smaller 60M/130M scales, FOSL remains competitive—within 1.18/0.69 PPL of Full-Rank—while retaining substantial savings (22–46% fewer parameters and 29–49% lower memory across scales); methods that keep all parameters but compress optimizer states (e.g., Fira) can occasionally edge out PPL in this regime. Overall, the accuracy gains of FOSL increase with model size, supporting our claim that folding with variance correction preserves wide-model quality at substantially reduced memory and parameter footprints.

**Equal-budget folding and foldable-only variants**. To isolate the effect of the folding ratio under a fixed parameter budget, we further sweep the folding ratio $\rho$ (and corresponding adapter rank $r$) for FOSL while matching the total parameter counts of LOST/SLTrain at each scale (43M, 94M, 185M, 609M). Tab. 3 reports validation PPL across LLaMA-60M/130M/350M/1B for several folding ratios and for a "foldable-only" variant that disables the low-rank path and trains only the folded sparse branch. Even without any low-rank component or additional optimization tricks, the foldable-only branch remains comparable with the full-rank baselines, while moderate folding ratios (e.g., $\rho = 0.8$–$0.9$) recover most of the accuracy at the same parameter budget.

**Ablation on mixing coefficient** $\gamma$. We reparameterize the mixing in Eq. 6 by a single coefficient $\gamma \in [0, 1]$ as $(\alpha, \beta) = (\gamma, 1 - \gamma)$. Following Li et al. (2025), we use a fixed $\gamma = 0.7$ by default. We ablate three variants on LLaMA-60M, 130M, 350M: (i) fixed $\gamma = 0.7$; (ii) a trainable per-layer scalar $\gamma$ with a bounded parameterization $\gamma = \sigma(\theta)$; and (iii) a trainable per-channel vector $\gamma \in [0, 1]^m$ applied at the addition. The obtained results are summarized in Tab. 4. Per-layer trainable $\gamma$ yields a small but consistent PPL reduction, while per-channel $\gamma$ degrades performance. We hypothesize that per-channel gating introduces many extra degrees of freedom, making optimization noisier and less well-conditioned.

| | 60M | | 130M | | 350M | | 1B | |
|---|---|---|---|---|---|---|---|---|
| Params budget (M) | 43 | | 94 | | 185 | | 609 | |
| | PPL↓ | Rank | PPL↓ | Rank | PPL↓ | Rank | PPL↓ | Rank |
| Full model | 30.27 | – | 23.13 | – | 18.76 | – | 16.52 | – |
| Low-rank-only | 35.13 | 128 | 26.71 | 256 | 21.77 | 256 | 18.22 | 512 |
| FOSL, $\rho$=0.99 | 33.21 | 127 | 24.38 | 251 | 18.77 | 249 | 14.91 | 499 |
| FOSL, $\rho$=0.90 | 32.57 | 98 | 24.14 | 207 | 18.53 | 191 | 14.88 | 382 |
| FOSL, $\rho$=0.80 | 32.42 | 66 | 23.93 | 159 | 18.37 | 126 | 14.99 | 253 |
| FOSL, $\rho$=0.70 | 32.46 | 34 | 23.71 | 110 | 18.46 | 61 | 15.20 | 123 |
| Foldable-only | 34.14 | 0 | 24.71 | 0 | 19.30 | 0 | 16.46 | 0 |

Table 3: Equal-parameter-budget ablation of FOSL across folding ratios $\rho$ (virtual-output sparsity) at fixed total parameter counts (43M, 94M, 185M, 609M). The "foldable-only" row removes the low-rank path and trains only the folded sparse branch.

| Setup | 60M PPL↓ | 130M PPL↓ | 350M PPL↓ |
|---|---|---|---|
| Fixed $\gamma$=0.7 | 31.61 | 23.95 | 18.52 |
| Trainable $\gamma$ (per-layer) | 31.45 | 23.82 | 18.23 |
| Trainable $\gamma$ (per-channel) | 31.58 | 23.72 | 18.50 |

Table 4: Ablation on mixing coefficient $\gamma$ for LLaMA-60M, 130M, and 350M (lower PPL is better). Default uses fixed $\gamma$=0.7 as in (Li et al., 2025). Trainable variants use a bounded parameterization $\gamma = \sigma(\theta)$. No LOST-style SVD initialization is used.

**Results on LLaMA 7B**. Due to space constraints, we report LLaMA-7B pre-training results for FOSL, full-rank Adam, 8-bit SLTrain, and LOST in Appendix E, including validation perplexity at multiple checkpoints together with batch size and measured memory per GPU.

**Initialization: default vs. LOST-style SVD**. Crucially, FOSL is trained from scratch and does not require warm-starting from a pre-trained full-rank checkpoint. By default, we initialize the low-rank adapter with $A$ drawn from Kaiming uniform (He et al., 2015a) and $B$ set to zeros; for the folded branch we initialize $W_{\text{base}}$ with Kaiming and build the reuse map $\pi$ using the balanced multi-permutation scheme in Sec. 3.2. Unless otherwise noted, this default is used for all main results. For completeness, we also evaluate a one-shot SVD-based initialization following Li et al. (2025). Starting from an initial dense weight $W^{(0)} \in \mathbb{R}^{m \times d}$, we compute the rank-$r$ truncated SVD $W^{(0)} \approx U_r \Sigma_r V_r^\top$ and set the adapter factors as $B \leftarrow U_r \Sigma_r^{1/2}$ and $A \leftarrow V_r \Sigma_r^{1/2}$ (rescaled to match our adapter convention). For the folded branch, we select the $m_{\text{base}}$ real rows of $W_{\text{base}}$ by top-$m_{\text{base}}$ residual energy from the SVD complement and copy those rows from $W^{(0)}$; the remaining (virtual) rows are assigned to their nearest real rows by cosine similarity to build $\pi$. As summarized in Tab. 5, SVD initialization yields consistent PPL improvements over the default, at the cost of an offline SVD pass.

Table 5: Comparison of default vs. LOST-style SVD initialization for LLaMA-60M, 130M and 350M with trainable layer-wise mixing coefficient (lower PPL is better). Unless otherwise noted, all main results use the default initialization.

| Setup | 60M PPL↓ | 130M PPL↓ | 350M PPL↓ |
|---|---|---|---|
| Default init (B:Zero, A:Kaiming) | 31.45 | 23.82 | 18.23 |
| LOST-style SVD init | 30.88 | 23.01 | 17.69 |

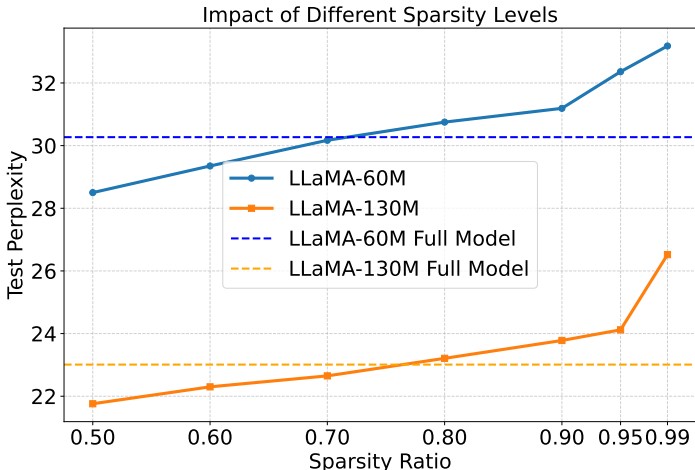

Figure 3: Impact of sparsity on test PPL for LLaMA-60M and LLaMA-130M. Dashed lines denote each model's dense (full) baseline. Lower is better.

| Model | ARC-C | | BoolQ | | HellaSwag | | MMLU | | PIQA | | Winogrande | |
| --- | --- | --- | --- | --- | --- | --- | --- | --- | --- | --- | --- | --- |
| | acc | acc_norm | acc | acc_norm | acc | acc_norm | acc | acc_norm | acc | acc_norm | acc | acc_norm |
| 1B Full-rank | 0.180 | 0.300 | 0.300 | – | 0.390 | 0.440 | 0.232 | – | 0.700 | 0.700 | 0.550 | – |
| 1B FOSL | 0.219 | 0.258 | 0.378 | – | 0.326 | 0.394 | 0.229 | – | 0.661 | 0.646 | 0.510 | – |

Table 6: Zero-shot downstream evaluation of LLaMA-1B full-rank vs. FOSL on a subset of `lm-evaluation-harness` tasks. Each entry reports accuracy / normalized accuracy (when provided by the harness; "–" otherwise).

**Impact of different sparsity**. We sweep the sparsity ratio $s \in \{0.50, 0.60, 0.70, 0.80, 0.90, 0.95, 0.99\}$ on LLaMA-60M and LLaMA-130M and report test perplexity (PPL) in Fig. 3. Two consistent trends emerge. First, PPL grows roughly monotonically with $s$: moderate sparsity (e.g., $s \leq 0.9$) incurs only a small decrease, while extreme sparsity ($s \geq 0.95$) triggers a sharp increase. Second, there exists a broad "safe" regime where FOSL remains close to (or even slightly better than) the dense baseline: for 60M, $s \approx 0.7$–$0.8$ matches the full model within $\sim 1$ PPL; for 130M, performance stays within $\sim 1$ PPL up to $s = 0.9$, but degrades markedly at $s = 0.99$. These results indicate that FOSL tolerates substantial sparsity before hitting a representation cliff, supporting our claim that channel reuse with variance correction preserves signal richness under tight compute budgets.

**Zero-shot downstream evaluation**. Beyond validation perplexity, we also evaluate zero-shot downstream performance of the LLaMA-1B full-rank and FOSL models on a subset of `lm-evaluation-harness` tasks. Tab. 6 reports accuracy (and normalized accuracy when available) on ARC-Challenge, BoolQ, HellaSwag, MMLU, PIQA, and Winogrande. FOSL tracks the full model closely across tasks: it improves over the dense baseline on ARC-Challenge and BoolQ, is slightly worse on HellaSwag, PIQA, and Winogrande, and essentially matches it on MMLU. Overall, these results indicate that the perplexity gains of FOSL at 1B translate into comparable zero-shot downstream performance under matched training budgets.

## 4.2 LLMs Fine-Tuning

To further assess the generalizability of FOSL, we fine-tune a pre-trained RoBERTa-base (Liu et al., 2019) on the GLUE benchmark (Wang et al., 2019), comparing against LoRA, GaLore, SLTrain, and LOST at ranks $r \in \{4, 8\}$ (LOST reports $r-1$ per its setup). Following (Han et al., 2024; Li et al., 2025), we parameterize each fine-tuned weight as $W + AB^\top + W_{fold}$, where $W$ denotes the full-rank pre-trained weights, $AB^\top$ is the low-rank adapter, and $W_{fold}$ is a foldable channel-wise

structured sparse matrix. We fine-tune the query and value projection layers while freezing the remaining parameters. As in LOST (Li et al., 2025), we remove the activation between the low-rank factors, which has negligible impact on these tasks.

As shown in Tab. 7, FOSL achieves performance comparable to strong low-rank baselines: it attains the best results on CoLA, SST-2, MNLI, and QNLI at $r = 4$, and the best MNLI (and a tie on SST-2) at $r = 8$. It is weaker on QQP, leading to slightly lower averages than the strongest baseline (LOST). This pattern is consistent with the fine-tuning regime, where only a small fraction of parameters and data are updated and standard LoRA-style adapters already perform strongly; consequently, gains over full-rank or vanilla LoRA are modest. The hyperparameters are summarized in Tab. 9.

Table 7: Performance comparison on GLUE (Wang et al., 2019) benchmark using RoBERTa-base (Liu et al., 2019) model with different low-rank training methods. Baseline results are reported from (Han et al., 2024; Li et al., 2025; Zhao et al., 2024).

| RoBERTa-base | CoLA | STS-B | MRPC | RTE | SST2 | MNLI | QNLI | QQP | Avg |
|---|---|---|---|---|---|---|---|---|---|
| Full-size | 62.24 | 90.92 | 91.30 | 79.42 | 94.57 | 87.18 | 92.33 | 92.28 | 86.28 |
| LoRA, $r = 4$ | 61.38 | 90.57 | 91.07 | 78.70 | 92.89 | 86.82 | 92.18 | **91.29** | 85.61 |
| Galore, $r = 4$ | 60.35 | 90.73 | 92.25 | 79.42 | 94.04 | 87.00 | 92.24 | 91.06 | 85.89 |
| LOST, $r = 3$ | 60.96 | **90.88** | **93.15** | **79.54** | 93.76 | 86.79 | 92.34 | 90.86 | 86.04 |
| **FOSL**, $r = 4$ | **61.94** | 90.10 | 91.90 | 76.90 | **94.73** | **87.16** | **92.55** | 86.51 | 85.22 |
| LoRA, $r = 8$ | 61.83 | 90.80 | 91.90 | 79.06 | 93.46 | 86.94 | 92.25 | 91.22 | 85.93 |
| Galore, $r = 8$ | 60.06 | 90.82 | 92.01 | **79.78** | 94.38 | 87.17 | 92.20 | 91.11 | 85.94 |
| SLTrain, $r = 8$ | 60.35 | 90.74 | 92.38 | 79.42 | 94.15 | 86.53 | 92.40 | **91.27** | 85.93 |
| LOST, $r = 7$ | **62.52** | **91.15** | 93.68 | 79.68 | 94.37 | 86.88 | **92.63** | 91.17 | 86.51 |
| **FOSL**, $r = 8$ | 61.59 | 90.35 | **92.39** | 78.34 | **94.38** | **87.33** | 91.98 | 86.96 | 85.39 |

## 5 CONCLUSION, LIMITATIONS AND OUTLOOK

We introduced FOSL, a foldable sparse-and-low-rank reparameterization that decouples true compute from interface width. By reusing channels via folding with lightweight variance correction, FOSL preserves dense tensor interfaces while paying the FLOPs of a narrower branch. The method integrates with standard training, is simple to implement, and yields favorable perplexity–memory / compute trade-offs compared against strong baselines.

**Limitations.** This work comes not without limitations. Variance correction assumes weak cross-channel correlations. Heavy-tailed or strongly dependent activations may reduce exactness. The reuse map and mixing coefficients are static in our current setup, limiting adaptation during training. Experiments focus on mid-scale LLMs and a narrow set of corpora due to compute constraints. Broader scales and tasks remain to be shown. Folding introduces gather / scatter patterns whose kernel- and communication-level efficiency is not yet fully optimized or profiled. Our theory gives conservative rank / expressivity statements, but tighter analyses under nonlinearities remain open.

**Outlook.** We plan to (i) learn the reuse map ($\pi$), learn per-layer sparsity, and add stabilized trainable per-output gains. Furthermore, (ii) we plan to scale FOSL to 7B+ models with multi-node training and co-designed gather / scatter kernels; (iii) extend FOSL to multimodal / vision / speech transformers while assessing robustness and transfer.

**Reproducibility Statement.**   We build on open-source implementations for efficient pre-training (GaLore (Zhao et al., 2024), ReLoRA (Lialin et al., 2023), SLTrain (Han et al., 2024)). We release an anonymized repository[1] with our FOSL code, and end-to-end scripts for data, training, and evaluation. Datasets, preprocessing / tokenization, sequence length, and environment / implementation details are in Appendix A, with per-scale hyperparameters in Tab. 8. Method and compute profile appear in Sec. 3 and Sec. 3.3. Baselines use identical protocols; token budgets and validation PPL are in Tab. 2, with ablations / sensitivity in Tab. 4 and Fig. 3. These materials and the anonymized repository enable independent reproduction of our results.

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

## APPENDIX

The following sections provide supplementary information omitted from the main text:

- Appendix D: Empirical evidence for variance correction.

## A  IMPLEMENTATION DETAILS

We trained over 100 models on a NVIDIA DGX Station A100 featuring eight NVIDIA A100 GPUs (each equipped with 80GB memory) to evaluate the performance of FOSL presented in this work. The training time vary from 45 minutes (LLaMA 60M) to 5 days (LLaMA 7B) depending on the model size. `Huggingface Hub`[2] is used to load the datasets. `WandB`[3] is used to log training history, training result, and evaluation metrics. The source code of all experiments is available at: `https://anonymous.4open.science/r/Efficient-Pretraining-preview-34C4/`

This section outlines the LLaMA architectures and the hyperparameters used during pre-training. To ensure fair comparison, we follow the same experimental settings as Zhao et al. (2024); Glentis et al. (2025). Tab. 8 summarizes the hyperparameters for different model scales. Across all architectures, we adopt a maximum sequence length of 256 and a batch size of 131,072 tokens. The learning rate is linearly warmed up during the first 10% of training steps, followed by a cosine annealing schedule that decays to 10% of the initial value. We use the T5-base tokenizer (Raffel et al., 2023), consistent with prior work (Han et al., 2024; Glentis et al., 2025).

Table 8: Pre-training hyperparameters of LLaMA architectures.

| Params | Hidden | Intermediate | Heads | Layers | Steps | Data (Tokens) |
|--------|--------|--------------|-------|--------|-------|---------------|
| 60M    | 512    | 1376         | 8     | 8      | 11K   | 1.3B          |
| 130M   | 768    | 2048         | 12    | 12     | 22K   | 2.6B          |
| 350M   | 1024   | 2736         | 16    | 24     | 65K   | 7.8B          |
| 1B     | 2048   | 5461         | 32    | 24     | 140K  | 13.1B         |

Table 9: Hyperparameters of FOSL for fine-tuning on GLUE (Wang et al., 2019). All models are trained with a fixed folding ratio $\rho = 0.9$.

|               | CoLA | | STS-B | | MRPC | | RTE | | SST-2 | | MNLI | | QNLI | | QQP | |
|               | r=4 | r=8 | r=4 | r=8 | r=4 | r=8 | r=4 | r=8 | r=4 | r=8 | r=4 | r=8 | r=4 | r=8 | r=4 | r=8 |
|---------------|-----|-----|-----|-----|-----|-----|-----|-----|-----|-----|-----|-----|-----|-----|-----|-----|
| Batch Size    | 16  | 16  | 16  | 16  | 16  | 16  | 16  | 16  | 16  | 16  | 16  | 16  | 16  | 16  | 16  | 16  |
| Epochs        | 30  | 30  | 30  | 30  | 30  | 30  | 30  | 30  | 10  | 10  | 15  | 15  | 10  | 10  | 15  | 15  |
| Learning Rate | 5e-5 | 5e-5 | 5e-5 | 5e-5 | 5e-5 | 5e-5 | 5e-5 | 5e-5 | 5e-5 | 5e-5 | 5e-5 | 5e-5 | 5e-5 | 5e-5 | 5e-5 | 5e-5 |
| $\alpha$      | 64  | 64  | 64  | 64  | 128 | 128 | 128 | 128 | 64  | 32  | tba | tba | 32  | 64  | 128 | 32  |

---

[2]https://huggingface.co/docs/hub/index

[3]https://wandb.ai

## B  FURTHER THEORETICAL RESULTS TO SUPPORT FOLDABLE AND SPARSE LOW-RANK PRE-TRAINING

Below we establish three key properties of the folded path. First, backpropagation through folding aggregates gradients from all virtual copies of a channel, effectively averaging them and leading to smoother updates. Second, the duplication operator is an isometry ($M^\top M = I$), which guarantees that gradients are not amplified and stability is preserved. Third, this aggregation improves the signal-to-noise ratio (SNR) of parameter updates: the effective signal grows with the number of copies, while the noise grows more slowly, yielding cleaner and more stable optimization dynamics. Together, these results explain why folding not only reduces computation, but also improves training stability compared to conventional sparse update methods.

Recall the folded path computes $z = W_{\text{base}} x \in \mathbb{R}^{m_{\text{base}}}$ and outputs $y_{\text{fold}} = D_\pi C z \in \mathbb{R}^m$, where the duplication map $\pi : \{1, \ldots, m\} \to \{1, \ldots, m_{\text{base}}\}$ induces disjoint groups $G_i = \{j : \pi(j) = i\}$ with sizes $|G_i| = 1 + k_i$. The diagonal correction $C = \text{diag}((1 + k_i)^{-1/2})$ stabilizes second moments under reuse.

**Backpropagation through folding.**  Let $g = \partial \mathcal{L} / \partial y_{\text{fold}} \in \mathbb{R}^m$. By the chain rule,

$$\frac{\partial \mathcal{L}}{\partial z} = C D_\pi^\top g \quad \Longrightarrow \quad \Big[\frac{\partial \mathcal{L}}{\partial z}\Big]_i = \frac{1}{\sqrt{1 + k_i}} \sum_{j \in G_i} g_j. \tag{13}$$

Therefore the row-wise gradient for $W_{\text{base}}$ is

$$\frac{\partial \mathcal{L}}{\partial W_{\text{base}}[i, :]} = \Big(\frac{1}{\sqrt{1 + k_i}} \sum_{j \in G_i} g_j\Big) x^\top, \tag{14}$$

which is a *scaled sum* of per-copy gradients sharing the same input $x$. Eq. 14 reveals a built-in averaging (aggregation) over copies. This shows that instead of treating each virtual copy independently, the folded path aggregates their contributions into a single update for the corresponding base channel. As a result, folding implicitly denoises gradients and provides more stable and efficient parameter updates without introducing additional computation.

**Isometry and stability.**  Let $M = D_\pi C \in \mathbb{R}^{m \times m_{\text{base}}}$. Its columns are normalized group indicators, hence $M^\top M = I_{m_{\text{base}}}$ (orthonormal columns). As a consequence, the folding map has operator norm $\|M\|_2 = 1$, and the adjoint $M^\top$ used in backprop does not amplify gradient norm (*i.e.*, $\|M^\top g\|_2 \leq \|g\|_2$). This provides a spectral-norm stability guarantee for gradient flow through the folded path. In other words, folding preserves the scale of gradient signals: while channels are reused to save computation, the backpropagated gradients remain well-conditioned, preventing exploding updates and ensuring stable optimization.

**Variance and SNR (general form).**  Let $g_{G_i} = (g_j)_{j \in G_i}$ denote the vector of gradients in group $G_i$, with mean $\mu_i = \mathbb{E}[g_j \mid j \in G_i]$ and covariance $\Sigma_i = \text{Cov}(g_{G_i})$. From Eq. 13, the aggregated scalar for base channel $i$ is

$$\Big[\frac{\partial \mathcal{L}}{\partial z}\Big]_i = \frac{1}{\sqrt{|G_i|}} \sum_{j \in G_i} g_j, \quad |G_i| = 1 + k_i.$$

Hence

$$\mathbb{E}\Big[\Big(\frac{\partial \mathcal{L}}{\partial z_i}\Big)\Big] = \sqrt{|G_i|}\, \mu_i, \qquad \text{Var}\Big[\Big(\frac{\partial \mathcal{L}}{\partial z_i}\Big)\Big] = \frac{1}{|G_i|} \mathbf{1}^\top \Sigma_i \mathbf{1}.$$

In particular, aggregation increases the *mean magnitude* by $\sqrt{|G_i|}$, while the variance depends on within-group covariance $\Sigma_i$. Thus, channel reuse amplifies the effective signal while not proportionally increasing the noise, so the signal-to-noise ratio of the gradients improves—explaining why folded updates are cleaner and training becomes more stable.

**Equicorrelated model and SNR gain.**  Under an equicorrelated noise model inside each group, $\Sigma_i = \sigma_i^2((1 - \rho_i)I + \rho_i \mathbf{1}\mathbf{1}^\top)$ with $\rho_i \in [-1, 1]$, we obtain

$$\text{Var}\Big[\Big(\frac{\partial \mathcal{L}}{\partial z_i}\Big)\Big] = \frac{1}{|G_i|} \sigma_i^2\Big((1 - \rho_i)|G_i| + \rho_i |G_i|^2\Big) = \sigma_i^2\big(1 + \rho_i k_i\big).$$

Compared to a single-copy gradient (variance $= \sigma_i^2$), the *variance itself* is unchanged when $\rho_i = 0$ (independent noise), and increases linearly with $\rho_i k_i$ when gradients are positively correlated. However, the *mean* scales as $\sqrt{|G_i|}\,\mu_i$, so the per-channel signal-to-noise ratio (SNR) improves by

$$\text{SNR gain} \;=\; \frac{\left(\sqrt{|G_i|}\,\|\mu_i\|\right)^2}{\sigma_i^2(1+\rho_i k_i)} \;\Big/\; \frac{\|\mu_i\|^2}{\sigma_i^2} \;=\; \frac{|G_i|}{1+\rho_i k_i} \;=\; \frac{1+k_i}{1+\rho_i k_i}.$$

Thus aggregation strictly improves SNR unless $\rho_i = 1$ (perfect correlation), and approaches a factor of $1 + k_i$ under independence ($\rho_i = 0$).

**Effect on parameter updates.** Using Eq. 14, the row-wise update for $W_{\text{base}}[i, :]$ is $\left(\frac{1}{\sqrt{|G_i|}}\sum_{j\in G_i} g_j\right)x^\top$. The update *mean* grows by $\sqrt{|G_i|}$ while the update variance scales as $\sigma_i^2(1+\rho_i k_i)\|x\|_2^2$ (per row). Consequently, the per-row SNR of $W_{\text{base}}$ updates improves by $\frac{1+k_i}{1+\rho_i k_i}$, which explains smoother loss trajectories and fewer spikes under the same optimizer/memory budget.

**Comparison to sparse residual updates.** In SLTrain (Han et al., 2024) / LOST (Li et al., 2025), the sparse residual $S$ updates element-wise (or per-structured block) without copy-wise aggregation. Let the set of active indices for row $i$ be $\Omega_i$ with $|\Omega_i| = k_i'$. The per-row gradient is a sum over *different inputs* $(x_j)$ at different coordinates, rather than over copies of the same scalar target, so the averaging structure in Eq. 14 does not arise and the variance reduction typically degrades to $\propto 1/k_i'$ only under strong independence and uniformity assumptions, while also suffering from sparse-kernel memory overheads. Empirically, FOSL's copy-wise aggregation behaves like an *effective batch-enlargement* along the base channels, leading to faster and more stable convergence under the same optimizer / memory budget.

**Remarks and edge cases.** (i) If within-group gradients are highly correlated ($\rho_i \approx 1$), SNR gains shrink. This motivates constructing $\pi$ such that grouped outputs tend to exhibit correlated *signals* but less correlated *noise* (randomized $\pi$ at init is a practical default). (ii) The isometry $M^\top M = I$ ensures no gradient amplification—any increase in mean step $\propto \sqrt{1 + k_i}$ is handled by the mixing $\beta$ and adaptive optimizers. (iii) Very large $k_i$ reduces $m_{\text{base}}$ and may underfit. In practice we choose $\rho$ to balance denoising benefits with base capacity.

## C  TRAINING DYNAMICS

To complement the perplexity and zero-shot results in the main text, we report training dynamics for the LLaMA-1B experiments. Fig. 4 shows the evaluation loss on the C4 validation split as a function of training step, comparing the full-rank 1B baseline to a FOSL variant with 609M trainable parameters under the same optimizer and schedule. The two curves closely track each other throughout training, with FOSL exhibiting slightly smoother trajectories and slightly lower final loss, consistent with the gradient-aggregation and denoising effects discussed in Appendix B. These dynamics support our claim that folding preserves stable optimization behavior even under aggressive channel reuse.

Going beyond the main 13.1B-token setting, we also train LLaMA-1B under a larger 26B-token budget motivated by Chinchilla-style scaling laws. Fig. 5 plots evaluation loss for the same full-rank vs. FOSL pair when both are trained to 26B tokens on C4. The curves remain well aligned over the longer horizon, with FOSL continuing to track the dense baseline and ending with a slightly higher validation loss. At the end of training on 26B tokens, the full-rank 1B model reaches validation PPL 13.44, while FOSL attains 14.77 under the same budget. This suggests that the benefits of folding are not an artifact of early stopping or under-training, and that FOSL remains stable in a more data-rich regime even when its final PPL is modestly worse than the dense model.

To evaluate smaller-scale behavior under prolonged training, we further run LLaMA-130M on a 100B-token C4 budget. Fig. 6 compares the evaluation loss of the full-rank 130M baseline and its FOSL counterpart with 94M parameters under the same optimizer and schedule. The two models exhibit similar trajectories, with FOSL ending up with a slightly higher final loss than the dense

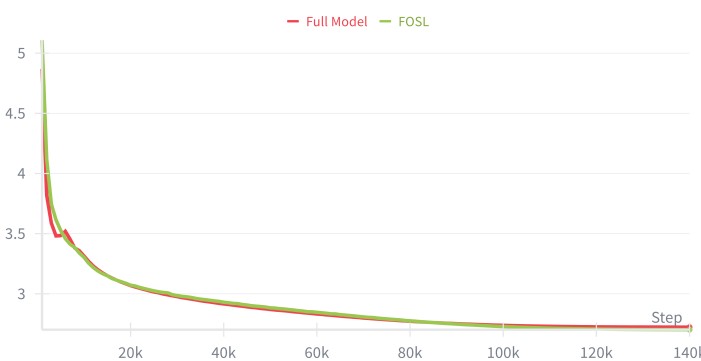

Figure 4: Evaluation loss vs. training step on C4 for the LLaMA-1B full-rank model and a FOSL variant with 609M parameters. Both runs use the same optimizer, schedule, and token budget (13.1B tokens). The curves largely overlap, indicating that FOSL matches the training stability of the dense model while achieving better final perplexity at a lower parameter and memory budget.

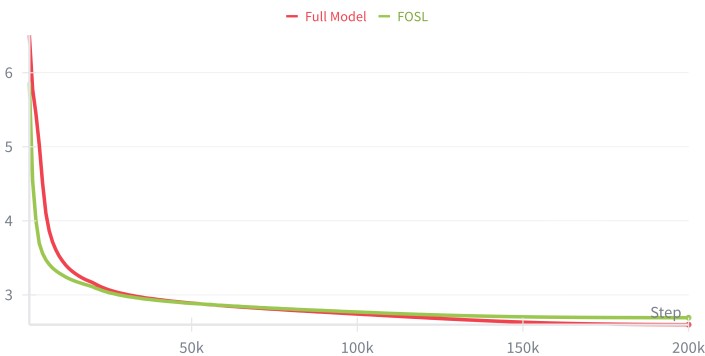

Figure 5: Evaluation loss vs. training step on C4 for LLaMA-1B full-rank and FOSL (same configuration as Fig. 4) when trained to a larger 26B-token budget. FOSL continues to match the dense model's training dynamics while achieving slightly lower final loss.

baseline. On this 100B-token run, the full-rank 130M model reaches validation PPL 18.73, whereas FOSL attains 20.72. Nevertheless, its performance remains reasonably close to the full model while benefiting from a smaller parameter and memory footprint. These results further support that folding does not introduce optimization instability even when models are trained far beyond the token budgets used in the main experiments.

## D EMPIRICAL EVIDENCE FOR VARIANCE CORRECTION

Our theoretical analysis in Appendix B shows that the diagonal rescaling $C_{ii} = (1+k_i)^{-1/2}$ prevents the variance of reused channels from blowing up when a real channel is copied $1 + k_i$ times by the duplication operator. Here we provide empirical evidence that (i) post-LayerNorm activations indeed exhibit weak cross-channel correlations, so a diagonal, second-moment correction is a reasonable approximation, and (ii) this correction materially improves training stability and final perplexity. Fig. 7a and Fig. 7b plot histograms of off-diagonal channel correlations for two representative FFN layers; in both cases the distribution is narrow and centered near zero, indicating that post-LN pre-activations are close to isotropic. Fig. 8 compares training curves for LLaMA-130M FOSL with and without variance correction and shows that removing the correction slows convergence and harms final loss, confirming its practical importance.

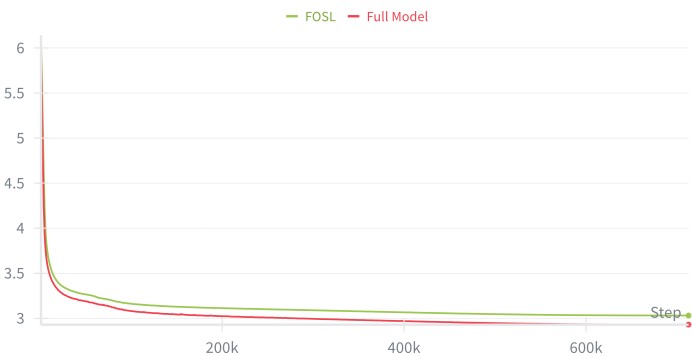

Figure 6: Evaluation loss vs. training step on C4 for LLaMA-130M full-rank and FOSL when trained on a 100B-token budget. FOSL closely follows the dense baseline and ends with a slightly higher but still comparable final loss while using fewer effective parameters and lower memory.

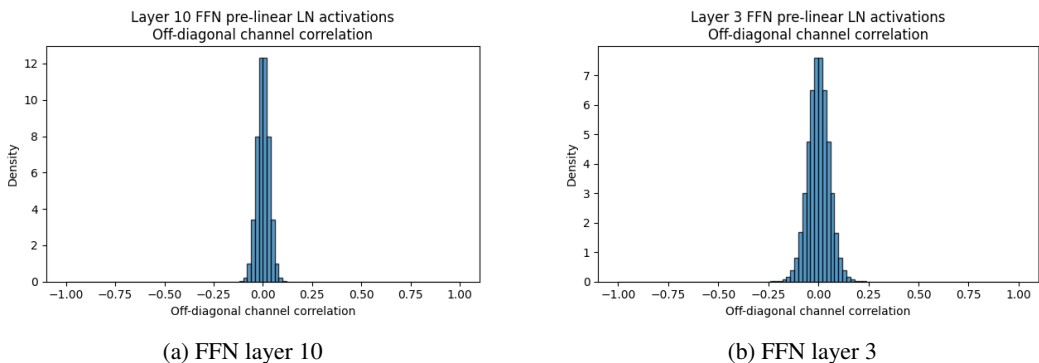

<table>
<tr><td>(a) FFN layer 10</td><td>(b) FFN layer 3</td></tr>
</table>

Figure 7: Off-diagonal channel correlations for two representative FFN layers (pre-linear LayerNorm activations). In both cases the distribution is narrow and centered near zero, indicating weak cross-channel correlations.

## E   LLAMA-7B RESULTS

For completeness, we also evaluate FOSL on LLaMA-7B pre-training on C4, following the 7B setup of SLTrain and LOST (Han et al., 2024; Li et al., 2025; Glentis et al., 2025). Tab. 10 reports validation perplexity at different training steps together with the effective batch size and peak memory per GPU. Full-rank Adam with batch size 4 cannot be pushed beyond 40K steps on our hardware, and 8-bit SLTrain with batch size 8 also stalls early. LOST scales to 150K steps with batch size 8, reaching 12.80 PPL, while FOSL uses batch size 8 at similar memory (61.25 GB vs. 62.15 GB) and achieves the best perplexity at later checkpoints (16.24 vs. 16.48 at 40K, 13.73 vs. 14.01 at 80K, and 12.50 vs. 12.80 at 150K). These results indicate that FOSL remains competitive at 7B scale and can better exploit larger batches under similar memory budgets.

## F   RUNTIME ANALYSIS

To address questions regarding the computational efficiency of FOSL, we measured the training throughput (tokens per second) and peak memory usage on a single node with 4 NVIDIA A100-80GB GPUs. Tab. 11 presents the results for LLaMA-350M and LLaMA-1B models. While FOSL introduces some overhead compared to the full-rank baseline due to the additional gather/scatter operations, it maintains comparable throughput.

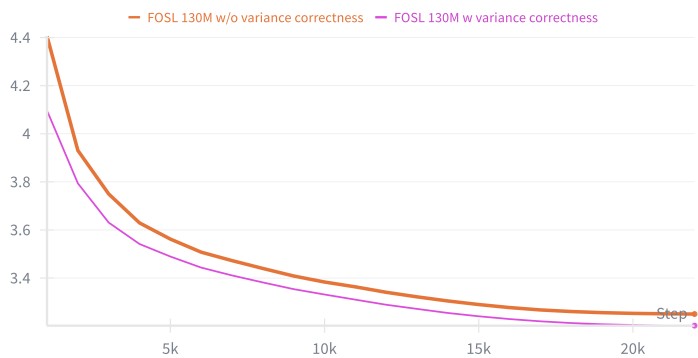

Figure 8: Evaluation loss vs. training step on C4 for LLaMA-130M FOSL with and without variance correction at a high folding ratio. The variance-corrected run converges faster and achieves lower final loss.

| Method | Batch size | Mem (G) | 10K | 40K | 80K | 120K | 150K |
|---|---|---|---|---|---|---|---|
| Full-Rank Adam | 4 | 49.53 | 24.95 | 20.05 | | N/A | |
| 8-bit SLTrain | 8 | 60.91 | 27.59 | | N/A | | |
| LOST | 8 | 62.15 | **24.41** | 16.48 | 14.01 | 12.93 | 12.80 |
| FOSL (ours) | 8 | 61.25 | 26.28 | **16.24** | **13.73** | **12.63** | **12.50** |

Table 10: Validation performance and actual memory footprint per GPU for the LLaMA-7B model pre-trained on C4 for up to 150K steps. Baseline results are taken from (Han et al., 2024; Li et al., 2025; Glentis et al., 2025); FOSL uses the same implementation and setup.

## G  RELATED WORK

Related work on efficient pre-training spans four complementary directions. First, low-rank adapters popularized by LoRA learn update-level low-rank corrections (Hu et al., 2021); pre-training variants such as ReLoRA and SwitchLoRA recover higher effective ranks via periodic merges or frequent subspace switching (Lialin et al., 2023; Zhou et al., 2025), while LORO directly optimizes a fixed-rank factorization on the low-rank manifold (Mo et al., 2025). Second, memory-efficient gradient projection methods (GaLore and variants) project gradients into low-rank subspaces to reduce optimizer state, with Fira stabilizing training via norm-based scaling and a norm-growth limiter (Zhao et al., 2024; Zhang et al., 2024; Chen et al., 2024). Third, sparse-plus-low-rank designs increase expressivity by adding sparse components: SLTrain employs a fixed unstructured support with low memory overhead (Han et al., 2024; Glentis et al., 2025), LOST co-designs complementary low-rank and channel-wise structured sparse components guided by an SVD initialization (Li et al., 2025), and CoLA inserts a nonlinearity between factors to enforce low-rank structure in activations (Liu et al., 2025). Similarly, FST (Hu et al., 2024) and SLoPe (Mozaffari et al., 2025) optimize training for semi-structured (e.g., 2:4) or block-sparse patterns to leverage specialized kernels. Finally, post-training model folding clusters redundant channels/heads and merges them with variance-preserving corrections to compress models without data or fine-tuning (Wang et al., 2025a; Jordan et al., 2022). FOSL brings the folding insight *into training*: it preserves full-width interfaces while reducing *true* compute by computing only $m_{\text{base}}$ real outputs and synthesizing the rest via a duplication operator with diagonal variance correction, which also aggregates copy-wise gradients (Appendix B) and yields denoising/SNR gains; compared to the above, this design reduces FLOPs and optimizer state like low-rank methods, avoids sparse-kernel overheads, and empirically delivers favorable perplexity–efficiency trade-offs across LLaMA scales under matched token budgets. Orthogonal to these four directions, a complementary line of work targets the activation-memory bottleneck by compressing activations stored for backpropagation: VeLoRA compresses intermediate activations via rank-1 sub-token projections and reconstructs them approximately during the backward pass (Miles et al., 2024), CompAct stores low-rank, randomly projected activations and only decompresses gradients for the optimizer update, achieving 25–30% peak-memory savings for LLaMA pre-training and up to 50%

Table 11: Runtime measurements on 4×A100-80GB GPUs. Throughput is measured in tokens per second, and Max Memory is the peak memory usage per GPU in GB.

| Model | Method | Throughput (tokens/s) | Max Memory (GB) |
|---|---|---|---|
| LLaMA-350M | Full-Rank | 442,331 | 33.40 |
| | SLTrain | 118,266 | 44.88 |
| | LOST | 259,031 | 53.44 |
| | FOSL | 194,509 | 52.88 |
| LLaMA-1B | Full-Rank | 105,627 | 24.67 |
| | SLTrain | 19,925 | 25.29 |
| | LOST | 78,886 | 25.52 |
| | FOSL | 61,650 | 32.82 |

for RoBERTa fine-tuning (Shamshoum et al., 2025), and tensor-decomposition methods compress activation maps using (high-order) SVD with theoretical guarantees on gradient approximation error (Nguyen et al., 2024).

# H  USE OF LARGE LANGUAGE MODELS

We primarily used ChatGPT[4] to correct grammatical errors in the manuscript and to fix minor compilation issues in Overleaf. In addition, Cursor[5] with GPT-5-High[6] was employed to debug programming errors encountered during implementation. Apart from these auxiliary uses, the research ideas, theoretical contributions, and the writing of this paper were entirely carried out by the authors. The code implementation is adapted from the open-source code of (Glentis et al., 2025).

---

[4]https://chatgpt.com/
[5]https://cursor.com/
[6]https://openai.com/index/introducing-gpt-5/

