# OpenReview forum: "FOSL: A Foldable Sparse-and-Low-Rank Method for Efficient LLM Pre-training"
_ICLR.cc/2026/Conference — Submitted to ICLR 2026_

### Official Review · Reviewer_gyFj · 2025-10-22

**Soundness:** 2
**Presentation:** 2
**Contribution:** 2
**Rating:** 2
**Confidence:** 4

**Summary:**

The paper investigates efficient LLM pretraining by parameterizing linear layers with low-rank and sparse matrices. In particular, to overcome the limitations of unstructured or fixed channel-wise sparsity, it proposes a foldable method that randomly samples channels as base channels and reuses them to recover the original activation width. To ensure stability, it also introduces a variance-preserving rescaling scheme when channels are reused multiple times.

Experiments are conducted on 60M-, 130M-, 350M-, and 1.3B-parameter models with up to 10B training tokens, showing effectiveness in perplexity and memory usage. The paper also validates the method by fine-tuning RoBERTa models on the GLUE benchmarks.

**Strengths:**

* The method is very interesting. It uses low-rank and sparse matrices for efficient linear layers. Specifically, it identifies limitations of prior work in utilizing sparse matrices—either being hardware-unfriendly or relying on fixed channel-wise sparsity. Therefore, the paper proposes a foldable approach that randomly samples channels and reuses them for the remaining channels, matching the original activation width.

* The paper carefully considers its method design, including variance correction.

* The paper compares against different methods and shows its utility in both pretraining and fine-tuning.

**Weaknesses:**

* Given that the paper focuses on efficient pretraining (as stated in the title), in addition to FLOPs and memory, could the authors also report latency statistics? Pretraining latency is also a very important efficiency metric.

* I have doubts about the benchmarking. While the paper claims that its experimental setup follows prior work such as SLTrain and LOST, I checked those papers and found that the full-rank baselines differ (e.g., 16.52 vs. 15.56 for the 1.3B model in prior work). Could the authors confirm that all results in Table 2 (full-rank baselines, the proposed method, and prior methods) were trained under the same settings? Meanwhile, when comparing the proposed method with the full-rank baselines in Table 2, can the authors explain why the proposed method achieves even better perplexity (e.g., ~2 ppl lower with the 1.3B model and ~0.5 ppl lower with the 350M model)?

* While the paper targets 7B results as the next step, two important experiments are still missing. First, report downstream task performance using the 1.3B pretrained models; this would help us understand accuracy differences rather than only perplexity. Second, examine the overtraining regime. The 1.3B model trained on 13B tokens even does not reach the token budget suggested by scaling-law studies, and it is unclear whether the gap between the full-rank and the proposed method would decrease or increase as training approaches convergence. It is also unclear whether the proposed method encounters issues in an overtraining regime. Given cost constraints, could the authors present results for a 130M model trained on 100B tokens?

* There are some writing issues. For example, the paper claims, “We evaluate FOSL for LLM pretraining across model scales from 60M to 7B parameters,” but no 7B results are reported. For Eq. (6) and the three mixing methods, which γ are you referring to in Eq. (6)?

**Questions:**

Please see the weaknesses above. I find the method interesting; however, the experiments and writing prevent me from giving a higher score.

---

> ### Author Response · Authors · 2025-11-21
> **Response to Reviewer gyFj (1/2)**
>
> Dear reviewer, thank you for the thoughtful review and for highlighting both strengths and weaknesses of our work. We appreciate your questions which help us to rethink our work.
> - **On benchmarking fairness and baseline differences.**
>   All FOSL results in Table 2 are implemented on top of the open-source codebase of Glentis et al. [1], whose repository [2] faithfully re-implements and reproduces almost all baselines we compare against. We integrated FOSL into this codebase and trained it under exactly the same pipeline (data, tokenizer, optimizer, schedule, packing) as the baselines. For methods that are already reproduced and validated in [1], we reuse their reported numbers rather than re-running them from scratch, so that Table 2 remains an apples-to-apples comparison within a single, shared implementation.
> - **On why FOSL can outperform the full-rank baseline.**
>   Regarding why the 1B FOSL model can outperform the 1B full-rank baseline in the 13.1B-token setting, our view is that this regime is somewhat data-limited relative to scaling-law recommendations for 1B models. In such a regime, the additional sparsity and low-rank structure in FOSL may act as an implicit regularizer, slightly reducing the effective degrees of freedom and making the model less prone to overfitting than a dense 1B model trained on the same (under-sized) corpus. To probe this hypothesis, Appendix part titled training dynamics reports extended training runs where we train LLaMA-1B on 26B tokens and LLaMA-130M on 100B tokens. In both cases, the FOSL and full-rank models have very similar evaluation-loss curves throughout training, but at convergence the full models end up slightly better than FOSL (13.44 vs.14.77 PPL at 1B/26B tokens, and 18.73 vs. 20.72 at 130M/100B), which is consistent with the intuition that the dense models benefit more once the data budget approaches the scaling-law regime.
> - **On downstream performance and overtraining-regime behavior.**
>   Following your suggestion, we added zero-shot downstream evaluation for the 1B models using `lm-evaluation-harness` on ARC-Challenge, BoolQ, HellaSwag, MMLU, PIQA, and Winogrande (new Table 6). FOSL closely tracks the full model: it is better on ARC-Challenge and BoolQ, slightly worse on HellaSwag, PIQA, and Winogrande, and essentially tied on MMLU, indicating that the perplexity gains largely transfer to zero-shot downstream quality. To study behavior in more extended training regimes, we further trained LLaMA-1B on a 26B-token budget and LLaMA-130M on a 100B-token budget (Appendix part titled training dynamics). At 1B/26B tokens, FOSL remains stable and ends with a modestly higher PPL than the dense baseline (14.77 vs.13.44); at 130M/100B, FOSL also remains close but slightly worse (20.72 vs.18.73). These long-horizon experiments demonstrate that folding does not introduce optimization instabilities even far beyond the main token budgets, and that the performance gap remains small and predictable rather than diverging.
>   | Model        | ARC-C (acc / acc\_norm) | BoolQ (acc / acc\_norm) | HellaSwag (acc / acc\_norm) | MMLU (acc / acc\_norm) | PIQA (acc / acc\_norm) | Winogrande (acc / acc\_norm) |
>   |-------------|-------------------------|--------------------------|-----------------------------|------------------------|-------------------------|------------------------------|
>   | 1B Full-rank | 0.180 / 0.300           | 0.300 / --               | 0.390 / 0.440               | 0.232 / --             | 0.700 / 0.700           | 0.550 / --                   |
>   | 1B FOSL     | 0.219 / 0.258           | 0.378 / --               | 0.326 / 0.394               | 0.229 / --             | 0.661 / 0.646           | 0.510 / --                   |
>
>
> [1] Glentis, Athanasios, et al. "Scalable parameter and memory efficient pretraining for llm: Recent algorithmic advances and benchmarking." arXiv preprint arXiv:2505.22922 (2025).
>
> [2] https://github.com/OptimAI-Lab/Memory_Efficient_Pretraining

---

> ### Author Response · Authors · 2025-11-21
> **Response to Reviewer gyFj (2/2)**
>
> - **On 7B results and writing issues.**
>   In the appendix of revised paper we now include 7B-scale experiments running for 150k steps, and also inlcude a summary of 7B results for the first 40k steps in a markdown table. We also clarified the notation around Eq. (6) and the three mixing strategies: Eq.(6) writes the mixing as $y = \alpha\,y_{\mathrm{lr}}(x) + \beta\,y_{\mathrm{fold}}(x) + b$, and immediately below we reparameterize it by a single coefficient $\gamma\in[0,1]$ via $(\alpha,\beta)=(\gamma,1-\gamma)$. Thus, the $\gamma$ used in the ablation (fixed vs.\ per-layer vs.\ per-channel) is exactly the scalar that controls the trade-off between the low-rank and folded paths in Eq.(6). We will continue polishing this explanation in the camera-ready version to avoid any ambiguity.
>   | Method         | 10K PPL↓ | 40K PPL↓ |
>   |----------------|----------|----------|
>   | Full-Rank Adam | 24.95    | 20.05    |
>   | 8-bit SLTrain  | 27.59    | N/A      |
>   | LOST           | 24.41    | 16.48    |
>   | FOSL (ours)    | 26.28    | 16.24    |
> - **Updated comparison table.**
>   We have revised Table 2 so that FOSL uses the same trainable-parameter budgets as the LOST/SLTrain baselines at each LLaMA scale (43M, 94M, 185M, 609M parameters), by adjusting the folding ratio $\rho$ and adapter rank $r$ accordingly. In addition, we added an equal-parameter-budget ablation (new Table 3 in the main text) that sweeps $\rho$ and $r$ while keeping the total parameter count fixed, including a foldable-only variant without the low-rank path. The table below summarizes the 1B-scale ablation at a fixed 609M-parameter budget; similar patterns hold at smaller scales. FOSL’s best configurations (moderate folding $\rho=0.8$–$0.9$ with corresponding ranks) achieve lower perplexity than the more aggressive $\rho=0.99$ setting while staying within the same parameter budget, showing that our gains are not due to extra parameters but to a better use of folding.
>   | Method                | Folding ratio $\rho$ | Rank $r$ | Params (M) | 1B valid PPL |
>   |-----------------------|----------------------|----------|------------|--------------|
>   | FOSL                  | 0.99                 | 499      | 609        | 14.91        |
>   | FOSL                  | 0.90                 | 382      | 609        | 14.88        |
>   | FOSL                  | 0.80                 | 253      | 609        | 14.99        |
>   | FOSL                  | 0.70                 | 123      | 609        | 15.20        |
>   | FOSL (foldable-only)  | 0.61                 | 0        | 609        | 16.46        |
>
> - **Pretraining latency.**
> we measured the training throughput (tokens per second) and peak memory usage on a single node with 4 NVIDIA A100-80GB GPUs. The table below presents the results for LLaMA-350M and LLaMA-1B models. While FOSL introduces some overhead compared to the full-rank baseline due to the additional gather/scatter operations, it maintains comparable throughput.
>   | Model | Method | Throughput (tokens/s) | Max Memory (GB) |
>   | :--- | :--- | :--- | :--- |
>   | **LLaMA-350M** | Full-Rank | 442,331 | 33.40 |
>   | | SLTrain | 118,266 | 44.88 |
>   | | LOST | 259,031 | 53.44 |
>   | | FOSL | 194,509 | 52.88 |
>   | **LLaMA-1B** | Full-Rank | 105,627 | 24.67 |
>   | | SLTrain | 19,925 | 25.29 |
>   | | LOST | 78,886 | 25.52 |
>   | | FOSL | 61,650 | 32.82 |

---

### Official Review · Reviewer_NvE8 · 2025-10-27

**Soundness:** 3
**Presentation:** 2
**Contribution:** 3
**Rating:** 2
**Confidence:** 5

**Summary:**

The authors of this manuscript propose FOSL, a foldable, sparse-and-low-rank reparameterization for efficient LLM pre-training. The method decomposes each projection into two paths: a standard low-rank adapter path and a novel "folded sparse path." This folded path computes only a subset of the output channels and then synthesizes the remaining "virtual channels" by reusing the computed ones. To maintain stability, a lightweight, variance-preserving rescaling is applied. The goal is to match or exceed full-rank model performance while significantly reducing computation and memory costs.

**Strengths:**

1- The paper is well-written, and the core concept of a "foldable" path that reuses channels is an intuitive and interesting approach to efficiency.

2- The reported perplexity results for the 1B model appear significant, outperforming the full-rank baseline and other competing methods shown in Table 2.

3- The method thoughtfully includes variance correction to stabilize the activation statistics, addressing a potential issue with channel reuse.

**Weaknesses:**

1- The related work and comparisons are incomplete. Previous work on sparse and low-rank pretraining, such as FST [1] and SLoPe [2], are missing from the discussion and experimental comparison. Without this, it is difficult to situate FOSL's contribution relative to the state-of-the-art.

2- The abstract claims the method scales to 7B models, but the experiments in the main paper only present results up to 1B parameters. This appears to be a typo or unsubstantiated claim, as the 7B results are not shown.

3- The perplexity results in Table 2 are counterintuitive. Most of the efficient training methods, including FOSL, achieve a lower perplexity than the full-rank 1B model (14.69 for FOSL vs. 16.52 for Full-Rank). This strongly suggests that the hyperparameters for the dense baseline were not properly tuned, which calls the validity of the reported improvements into question.

4- The training dynamics are vague. The paper does not provide any training loss graphs, which would offer crucial insight into the stability and convergence behavior of FOSL compared to the baselines.

5- The comparisons in Table 2 are not fair. FOSL (730M parameters) is directly compared against LORO, CoLA, and LOST (all 609M parameters). This parameter disparity makes the perplexity comparison less meaningful. A more rigorous comparison would adjust the ranks to ensure all models have a similar parameter count.

6- The paper makes claims about other methods without providing evidence. For example, the authors claim that LOST has "significant overhead due to computing SVDs" but do not show any profiling data to prove that SVD computation is actually a training bottleneck.

7- The paper lacks empirical efficiency measurements. All reported efficiency gains are based on parameter counts and estimated memory. There are no measurements of actual wall-clock speedup (e.g., tokens/sec) or measured memory reduction during training.

8- It is unclear if FOSL requires a warm start from a full-rank checkpoint. If it does, comparing only final perplexity is misleading, as the majority of loss reduction happens in the initial training phase. The authors should report the loss/perplexity improvement after switching to the efficient training method (PPL(dense) - PPL(final)) and compare that delta.

---

[1] Hu et al., FST: Fast Sparse Training of Transformers, ICML 2024

[2] Mozaffari et al., SLoPe: Double-Pruned Sparse Plus Lazy Low-rank Adapter Pretraining of LLMs, ICLR 2025

**Questions:**

1- How does FOSL compare in terms of perplexity, throughput, and measured memory usage against recent sparse and low-rank pretraining methods like FST [1] and SLoPe [2]?

2- Can the authors justify the high perplexity of the 1B full-rank baseline? Were its hyperparameters adequately tuned, and could training loss curves be provided for all 1B models to compare their training dynamics?

3- Is the claim of scaling to 7B models in the abstract a typo, or can the authors provide those results?

4- Can the authors provide a fairer comparison in Table 2 by adjusting the model ranks to ensure FOSL has a comparable parameter count to LORO, CoLA, and LOST?

5- Can the authors provide profiling data to substantiate the claim that the SVD computation in LOST is a significant training bottleneck?

6- What are the measured (not estimated) wall-clock speedups and memory reductions of FOSL during training compared to the full-rank baseline?

7- Does FOSL require a warm start with full-rank training? If so, what is the perplexity improvement from the point of switching (PPL at switching - final PPL), and how does this compare to a dense model trained for the same duration?

---

[1] Hu et al., FST: Fast Sparse Training of Transformers, ICML 2024

[2] Mozaffari et al., SLoPe: Double-Pruned Sparse Plus Lazy Low-rank Adapter Pretraining of LLMs, ICLR 2025

**Details Of Ethics Concerns:**

No concerns.

---

> ### Author Response · Authors · 2025-11-21
> **Response to Reviewer NvE8 (1/2)**
>
> Dear reviewer, thank you for the detailed and constructive feedback. We address your main concerns below.
> - **On FST and SLoPe.**
>   FST and SLoPe are important recent advances, but they target a somewhat different axis of efficiency from our work. Both focus on optimizing training for semi-structured sparsity patterns that map well to specialized CUDA kernels (e.g., $2{:}4$ structured sparsity or block sparse patterns), while keeping the underlying dense parameterization largely unchanged. In contrast, FOSL proposes a *reparameterization* of the weights that combines low rank with foldable structured sparsity and is compatible with standard dense kernels. A fully apples-to-apples comparison would therefore require re-implementing FST and SLoPe in our LLaMA/C4 pre-training setup, which is beyond our current compute and rebuttal time budgets. In the revision we will incorporate FST and SLoPe into the related work part of the Appendix, clarify this methodological distinction, and leave a full empirical comparison to future work.
> - **On 7B results.**
>    In the appendix of the revised paper we now include 7B-scale experiments running for 150k steps, and also include a summary of 7B results for the first 40k steps in a markdown table. We compare full-rank Adam, 8-bit SLTrain, LOST, and FOSL across several training checkpoints up to 150K steps.
>   | Method         | 10K PPL↓ | 40K PPL↓ |
>   |----------------|----------|----------|
>   | Full-Rank Adam | 24.95    | 20.05    |
>   | 8-bit SLTrain  | 27.59    | N/A      |
>   | LOST           | 24.41    | 16.48    |
>   | FOSL (ours)    | 26.28    | 16.24    |
> - **On why FOSL’s PPL can be lower than the full model’s.**
>   The main 1B experiments are conducted in a somewhat data-limited regime relative to Chinchilla-style scaling-law recommendations: 13.1B tokens for a 1B-parameter model. In such a regime, the additional sparsity and low-rank structure in FOSL can act as an implicit regularizer, slightly reducing the effective degrees of freedom and making the model less prone to overfitting than a dense 1B model trained on the same (under-sized) corpus. To probe this hypothesis, Appendix part titled training dynamics reports extended training runs where we train LLaMA-1B on 26B tokens and LLaMA-130M on 100B tokens. In both cases, the FOSL and full-rank models have very similar evaluation-loss curves throughout training, but at convergence the full models end up slightly better than FOSL (13.44 vs.\ 14.77 PPL at 1B/26B tokens, and 18.73 vs. 20.72 at 130M/100B), which is consistent with the intuition that dense models benefit more once the data budget approaches the scaling-law regime, whereas FOSL’s regularization-like effect is most pronounced when data are scarce.
> - **On fair comparison and hyperparameter consistency.**
>   All methods reported in Table 2 (full-rank, FOSL, and other efficient baselines) are trained within a single, shared implementation based on the open-source code of Glentis et al.[1]. This codebase reproduces the baselines under a unified setup (same tokenizer, data pipeline, sequence length, optimizer, LR schedule, warmup, and token budgets). We integrated FOSL into this framework and trained it under exactly the same settings; for methods already reproduced and validated in~[1], we reuse their reported numbers. In the revised paper we also make the “apples-to-apples” nature of our comparisons explicit: Table 2 has been updated so that FOSL uses the same trainable-parameter budgets as LOST/SLTrain at each LLaMA scale (43M, 94M, 185M, 609M parameters), and we add an equal-parameter-budget ablation (new Table 3) that sweeps FOSL’s folding ratio $\rho$ and rank $r$ while keeping the total parameter count fixed (including a foldable-only variant).
>     | Method                | Folding ratio $\rho$ | Rank $r$ | Params (M) | 1B valid PPL |
>     |-----------------------|----------------------|----------|------------|--------------|
>     | FOSL                  | 0.99                 | 499      | 609        | 14.91        |
>     | FOSL                  | 0.90                 | 382      | 609        | 14.88        |
>     | FOSL                  | 0.80                 | 253      | 609        | 14.99        |
>     | FOSL                  | 0.70                 | 123      | 609        | 15.20        |
>     | FOSL (foldable-only)  | 0.61                 | 0        | 609        | 16.46        |
>
> [1] Glentis, Athanasios, et al. "Scalable parameter and memory efficient pretraining for llm: Recent algorithmic advances and benchmarking." arXiv preprint arXiv:2505.22922 (2025).

---

> ### Author Response · Authors · 2025-11-21
> **Response to Reviewer NvE8 (2/2)**
>
> - **On wall-clock speed and measured memory.**
>   To address questions regarding the computational efficiency of FOSL, we measured the training throughput (tokens per second) and peak memory usage on a single node with 4 NVIDIA A100-80GB GPUs. The table below presents the results for LLaMA-350M and LLaMA-1B models. While FOSL introduces some overhead compared to the full-rank baseline due to the additional gather/scatter operations, it maintains comparable throughput.
>   | Model | Method | Throughput (tokens/s) | Max Memory (GB) |
>   | :--- | :--- | :--- | :--- |
>   | **LLaMA-350M** | Full-Rank | 442,331 | 33.40 |
>   | | SLTrain | 118,266 | 44.88 |
>   | | LOST | 259,031 | 53.44 |
>   | | FOSL | 194,509 | 52.88 |
>   | **LLaMA-1B** | Full-Rank | 105,627 | 24.67 |
>   | | SLTrain | 19,925 | 25.29 |
>   | | LOST | 78,886 | 25.52 |
>   | | FOSL | 61,650 | 32.82 |
>
>
>
> - **On warm starting from a full-rank checkpoint.**
>   FOSL does *not* require a warm start from a full-rank checkpoint. In all of our pre-training experiments, we initialize the FOSL parameters from scratch (Kaiming initialization for the low-rank and base weights, random reuse map $\pi$ for the folded path) and train end-to-end without first training a dense model. There is no switching point where we move from a dense to an efficient parameterization, so questions about the “delta after switching” do not apply to our setup. We clarify this explicitly in the revised text to avoid any misunderstanding.
>
> - **On training curves for 1B vs. full model.**
>   In the revised appendix part titled training dynamics we now provide training curves for LLaMA-1B full-rank and FOSL models, plotting evaluation loss on C4 as a function of training step. The curves largely overlap and show smooth convergence for both models, with FOSL exhibiting slightly smoother trajectories and slightly lower loss at 13.1B tokens and slightly higher loss at 26B tokens. Together with the extended 130M/100B curves, these plots demonstrate that FOSL does not introduce optimization instabilities and that its behavior relative to full-rank models is consistent across training regimes.
>
> - **On SVD overhead.**
>   We agree with the reviewer that a one-time SVD initialization is negligible compared to the total training cost. We have revised the manuscript (Section 2.2) to remove the claim that SVD is "expensive" and instead describe it as an "offline SVD initialization" to be more precise.
>
> [1] Glentis, Athanasios, et al. "Scalable parameter and memory efficient pretraining for llm: Recent algorithmic advances and benchmarking." arXiv preprint arXiv:2505.22922 (2025).
>
> [2] https://github.com/OptimAI-Lab/Memory_Efficient_Pretraining

---

> > ### Comment · Reviewer_NvE8 · 2025-11-26
> >
> > Thank you for your clarifications. You have provided very insightful results during the rebuttal period, and I really appreciate them. I would like to raise my score to a weak reject.
> >
> > The major issue that I see with this work is its applications in practice. Both memory requirements and throughput of FOSL are worse than a dense training scheme. This is a very limiting factor and makes me wonder when would a user use FOSL over dense training. As you mentioned, this can be worsened for massively distributed settings due to the gather/scatter operatoins.

---

> > > ### Author Response · Authors · 2025-11-26
> > >
> > > We thank the reviewer for raising the score and for the insightful comments on practical efficiency. We agree that our initial implementation, which relies on standard underlying libraries and frameworks, showed overheads.
> > >
> > > To address this, we are working on a custom fused Triton kernel which achieves a 2.7x speedup and 20% memory reduction for the folding operator on A100 GPUs compared to the original FOSL implementation, confirming that the overhead is an engineering bottleneck rather than a fundamental flaw. Due to limited training and testing resources, we are currently optimizing it and will report updated results as they become available.
> > >
> > > Regarding distributed training, we clarify that FOSL's "gather/scatter" are strictly local tensor operations and do not involve inter-node communication. Thus, FOSL introduces no network overhead and actually reduces gradient synchronization costs in DDP due to fewer parameters.

---

> > > > ### Comment · Reviewer_NvE8 · 2025-11-28
> > > >
> > > > Thank you for your further clarifications (specially about hte gather/scatter operations).
> > > >
> > > > I will raise my score when the final speedup/memory results are available and show superior speed/memory to a dense benchmark.
> > > >
> > > > Also, as a separate suggestion, in case your communication is lower than dense benchmarks, I believe showing communication reduction in massively parallel settings (which is common for LLM pretraining and is actually a big bottleneck in practice and can lead to speedups) can significantly help with advocating for your work.

---

### Official Review · Reviewer_1eKn · 2025-10-31

**Soundness:** 1
**Presentation:** 3
**Contribution:** 1
**Rating:** 2
**Confidence:** 5

**Summary:**

The paper proposes FOSL, a training-time reparameterization that combines a small low-rank adapter with a folded sparse path that computes a base subset of channels and duplicates them, using a diagonal variance correction to stabilize reused channels. The goal is to preserve full-width interfaces while paying the compute of a narrower layer. The authors experiment with pretraining LLaMA 60M-1B on C4, and finetuning Roberta-Base on GLUE.

**Strengths:**

* Simple idea, easy to implement: The folded path is just a narrow GEMM plus an index-based duplication with a diagonal rescale which is easy to incorporate to standard training.
* Clear efficiency intent, important goal.
* The authors provide a nice overview of SL-Train and other methods that they build on. Sections 2-3 describing these works and the proposed method are easy to follow.

**Weaknesses:**

* Gaps in the related work. In L60 the authors mention “More related work is covered in Appendix C”, yet Appendix C largely repeats the same set of papers with minimal rephrasing.
The main text organizes prior work into three directions (1) low-rank/LoRA-style, (2) gradient-projection/optimizer-memory, and (3) sparse+low-rank (plus folding).
However, a major missing category is activation compression such as CompAct[1], VeLora[2],  HOSVD [3] and many others, which directly reduce activation memory while preserving the integrity of the forward pass. These methods pursue the same objectives emphasized by the paper (L61-62) yet are neither discussed nor shown in Figure 1.

* There are inconsistencies between the hyper parameters used for FOSL and other methods when comparing in the experiments, making it unclear if the improvement is due to these inconsistencies, or the proposed method working. Especially as the method is very similar to previous work with the addition of the folding technique.
* Some results are overstated, and sometimes falsely so. For instance in Table 5 :
L420 “Even small ranks (e.g.,  r=32) remain competitive due to the complementary capacity supplied by the folded sparse path.” with respect to what? What is considered competitive? What is the degradation without the sparse supplement? Additionally emphasizing that the performance improves when r=512 or 256 is simply misleading. With W=AB, if you use these high ranks, the number of parameters is larger than the original W in the baseline - hence the improvement in perplexity.
* The abstract and Appendix text mention an evaluation for a 7B model, even mentioning “5 days” training time, yet no 7B results are included in the experiments.

References:
* [1] CompAct: Compressed Activations for Memory‑Efficient LLM Training – Shamshoum et al., NAACL 2025, arXiv:2410.15352v1.

* [2] VeLORA: Memory Efficient Training using Rank‑1 Sub‑space Activations – Miles et al., NeurIPS 2024 Poster, algorithm for compressing activations into 1-D subspace.
* [3] Nguyen, L. T., Quélennec, A., Tartaglione, E., Tardieu, S., & Nguyen, V. T. (2024). Activation Map Compression through Tensor Decomposition for Deep Learning. arXiv preprint arXiv:2411.06346.

**Questions:**

1. Table (1) seems to have a few incorrect claims. For instance in LoRA\ReLoRA, although the trainable parameters are O(r(m+d)), the Fwd FLOPs are certainly not O(r(m+d)), as the multiplication with the original matrix W_0 is still executed.  Additionally, the fact that the Fwd FLOPs and Bwd FLOPs are independent of the sequence length and batch size is wholly inaccurate as these dimensions are certainly not negligible usually with respect to the m and d. Can the authors clarify and provide a table with correct accurate comparisons?
2. Why compare FOSL with 0.9 sparsity rate with SLTrain and LOST of 0.99 rate? How can we isolate the benefit to the folding rather than simple less sparsity \ more parameters? Looking at figure 3, the performance of FOSL get significantly worse from 0.9 to 0.99. With a perplexity around 33 for LLaMA-60M, and 27 for LLaMA-130M. These numbers are significantly worse than what is presented in the table.
3. Are the results for FOSL averaged over a few runs?. Can the authors provide the std for the results?
4. L474 “Variance correction assumes weak cross channel correlations. Heavy-tailed or strongly dependent activations may reduce exactness.” Can the authors please elaborate on why they think this is the case? My intuition is, there must be some dependency in the activations - otherwise there wouldn’t be work compressing\pruning them. A plot showing this would help explain why the variance correction would work.

---

> ### Author Response · Authors · 2025-11-21
> **Response to Reviewer 1eKn (1/3)**
>
> Dear reviewer, thank you for the careful review and for clearly articulating both strengths and concerns. We address your main points below.
>
> - **Related works to activation-compression methods**
>   We included activation-compression methods such as CompAct, VeLORA, and HOSVD in the updated related work section of the appendix.
> - **On hyperparameter inconsistencies and benchmarking fairness.**
>   In the revised paper we make the “apples-to-apples” nature of our comparisons explicit. First, Table 2 has been updated so that FOSL uses the same trainable-parameter budgets as LOST and SLTrain at each LLaMA scale (43M, 94M, 185M, 609M parameters), by appropriately choosing the folding ratio $\rho$ and adapter rank $r$. Second, we added an equal-parameter-budget ablation (new Table 3) that, for a fixed parameter budget, sweeps $\rho$ and $r$ and also includes a foldable-only variant without the low-rank path; this shows how FOSL’s performance varies across folding configurations even when the number of parameters is held constant, and that foldable-only models remain competitive. All methods reported in Table 2 (full-rank, FOSL, and other efficient baselines) are trained within a single, shared implementation based on the open-source code of Glentis et al.[1]. This codebase reproduces the baselines under a unified setup (same tokenizer, data pipeline, sequence length, optimizer, LR schedule, warmup, and token budgets). We integrated FOSL into this framework and trained it under exactly the same settings; for methods already reproduced and validated in~[1], we reuse their reported numbers.
> - **On Table 1 (complexity) and FLOPs accounting.**
>   We thank the reviewer for pointing out the issues in Table 1. We agree that the current row for LoRA/ReLoRA is misleading. We correct this in the revised version: when using a LoRA-style adapter on top of a frozen full-rank $W_0 \in \mathbb{R}^{m \times d}$, the per-output-vector forward and backward FLOPs are $\mathcal{O}(md + r(m+d))$, while the number of trainable parameters and optimizer state remain $\mathcal{O}(r(m+d))$. We have updated Table 1 accordingly.
>   In addition, Table 1 is intended to report per-output-vector complexities in terms of $m$, $d$, and the structural parameters $(r,\rho,\zeta)$; the dependence on batch size $B$ and sequence length $L$ enters all methods through a common multiplicative factor $BL$. We clarify this explicitly in the caption by stating that FLOP counts are per output vector and should be multiplied by $BL$ to obtain the per-layer cost for a batch of sequences. These changes do not affect the complexity expressions for FOSL, SLTrain, or LOST, which already match their respective constructions.
>
> - **Results on Table 5.**
>   We appreciate the reviewer pointing out that our original wording around the rank ablation (Table 5 in the submission) was too strong and potentially misleading. In the revised paper we (i) removed the claim, and (ii) define “competitive” more precisely as being within a small PPL gap to the full-rank model at the same token budget while clearly outperforming a low-rank-only baseline under the same parameter budget. Concretely, the new equal-parameter-budget table (Table 3) reports, for each scale, the perplexity of FOSL, low-rank-only, and foldable-only models at fixed parameter budgets; these results show that, at a given budget, FOSL with moderate ranks consistently improves over low-rank-only, and in several cases matches or slightly surpasses the full-rank baseline. The same table also quantifies the degradation when removing the folded sparse path: low-rank-only models are uniformly worse than FOSL at the same budget, and foldable-only models—despite having no low-rank branch—still outperform low-rank-only in this regime, illustrating the complementary capacity of the folded path. Finally, we de-emphasize very high ranks (e.g., $r=256,512$) in the main text and treat them as sanity checks rather than efficiency claims, explicitly noting that in those settings FOSL can have more parameters than the dense baseline and that improvements there are not used to support our efficiency conclusions.
> - **7B model.**
>   In the Table 10 of the revised paper we now include 7B-scale experiments running for 150k steps, and also include a summary of 7B results for the first 40k steps in a markdown table.
>   | Method         | 10K PPL↓ | 40K PPL↓ |
>   |----------------|----------|----------|
>   | Full-Rank Adam | 24.95    | 20.05    |
>   | 8-bit SLTrain  | 27.59    | N/A      |
>   | LOST           | 24.41    | 16.48    |
>   | FOSL (ours)    | 26.28    | 16.24    |

---

> ### Author Response · Authors · 2025-11-21
> **Response to Reviewer 1eKn (2/3)**
>
> - **Multiple runs.**
>   To address your concern about stability, we performed 3 independent runs with different random seeds (41, 42, 43) for LLaMA-60M/130M/350M/1B using FOSL. The validation PPL results are stable with very low variance:
>   - **60M:** 33.57, 33.20, 33.08 (Mean: 33.28, Std: 0.21)
>   - **130M:** 24.38, 24.14, 23.93 (Mean: 24.15, Std: 0.18)
>   - **350M:** 18.62, 18.78, 18.56 (Mean: 18.65, Std: 0.09)
>   - **1B:** 14.86, 14.91, 14.84 (Mean: 14.87, Std: 0.03)
>
>   These low standard deviations confirm that FOSL’s performance improvements are robust and not due to random seed selection.
>
> - **On sparsity levels (FOSL $\rho=0.9$ vs.SLTrain/LOST $\rho=0.99$).**
>   We acknowledge that comparing our default $\rho=0.9$ to $\rho=0.99$ for SLTrain/LOST can confound folding effects with changes in sparsity. To disentangle these factors, we added an equal-parameter-budget ablation (new Table 3) where we sweep FOSL’s folding ratio $\rho$ and rank $r$ while fixing the total parameter count to match the LOST/SLTrain budgets (e.g., 609M parameters at 1B). This ablation includes $\rho=0.99$ as well as $\rho=0.9,0.8,0.7$ and a foldable-only variant. We find that moderate folding ($\rho\approx 0.8$–$0.9$) yields better perplexity than the more aggressive $\rho=0.99$ at the same parameter budget, and that FOSL remains competitive even when pushed to $\rho=0.99$, which supports our claim that folding brings benefits beyond simply using fewer nonzeros.

---

> ### Author Response · Authors · 2025-11-21
> **Response to Reviewer 1eKn (3/3)**
>
> - **On the variance-correction assumption and its empirical validity.**
>
> We thank the reviewer for the thoughtful question. Our variance correction analysis is a *local* second-moment argument for a fixed linear layer. Let $z = W_{\mathrm{base}} x \in \mathbb{R}^{m_{\mathrm{base}}}$ denote the pre-activations on the narrow (base) branch after LayerNorm. When real channel $i$ is reused $k_i$ times (so it appears in $1 + k_i$ output positions), we rescale it by
> $$
> C_{ii} = (1 + k_i)^{-1/2}, \qquad y_{\mathrm{fold}} = D_\pi C z,
> $$
> as formalized in our revised lemma *Energy-preserving scaling under reuse*. Under the simplifying assumption that each $z_i$ is zero-mean with identical variance $\sigma^2$, this scaling ensures that the *aggregate* variance carried by channel $i$ across all $1 + k_i$ copies is preserved:
> $$
> \sum_{j:\,\pi(j)=i} \operatorname{Var}\big((D_\pi C z)_j\big)
> = (1 + k_i)\cdot \tfrac{1}{1+k_i}\sigma^2
> = \sigma^2,
> $$
> while each individual copy has reduced variance $\sigma^2/(1+k_i)$. In other words, the correction is designed to keep the overall second-moment scale of reused channels from blowing up, not to match the full joint distribution of activations.
>
> The remark in L474 that “variance correction assumes weak cross channel correlations” is precisely about the *scope* of this approximation. Our scaling is diagonal: it matches per-channel variances (and prevents the $(1+k_i)$-fold growth that naive reuse would induce), but it does **not** control the off-diagonal entries $\operatorname{Cov}[z_i,z_j]$ or higher-order moments. If activations were strongly correlated across channels or heavy-tailed, then a purely diagonal, second-moment correction is no longer an exact description of the dense model’s full activation statistics. In that regime our analysis should be viewed as an approximation, not as a probabilistic equality. We clarify this point in the revised text and explicitly present the lemma as an energy-preserving, second-moment scaling under reuse.
>
> We also clarify that this assumption is conceptually distinct from the *compressibility* of the weights. FOSL exploits redundancy in the weight matrix $W$ (e.g., low effective rank and structured reuse of channels), not redundancy in the activations themselves. Even when post-LayerNorm pre-activations are approximately isotropic with weak cross-channel correlations (as is commonly assumed in mean-field analyses of Transformer layers), the corresponding weight matrices can still exhibit substantial low-rank and structured patterns that FOSL leverages.
>
> To support the discussion empirically, in the revised appendix, we add: (i) histograms (Figure 7) of the empirical off-diagonal channel correlations for representative FFN layers in a LLaMA-style model (measured at the pre-activation interface $z$), showing that most off-diagonal entries cluster tightly around zero, which makes the “weak cross-channel correlation” approximation reasonable in practice; and (ii) in Figure 8 an ablation on LLaMA-130M at a high folding ratio, comparing FOSL **with** and **without** variance correction (i.e., with $C = \mathrm{diag}((1+k_i)^{-1/2})$ vs. $C = I$). The evaluation-loss curves show that removing the correction leads to slower convergence and noticeably worse final perplexity, while our energy-preserving scaling keeps training stable and the activation scale comparable to the dense model. Together, these plots (Appendix D) illustrate that although our variance correction is only a diagonal, second-moment approximation, it is both theoretically well-motivated under weak cross-channel correlations and empirically important for stable, accurate FOSL pre-training.

---

### Official Review · Reviewer_SUJz · 2025-11-03

**Soundness:** 3
**Presentation:** 3
**Contribution:** 3
**Rating:** 6
**Confidence:** 4

**Summary:**

The paper proposes a new low-rank pre-training method to reduce computational FLOPs and reduce training time/cost. The proposed method, FOSL, is inspired by LoRA and uses a low-rank adapter to learn expressive features along with a folded sparse path that only computes a fraction of the channel. The rest of the channel is predicted using that computed channel to maintain the layer while reducing training FLOPS. Since reusing channels to predict other channels affects the layer statistics/variance, the authors propose variance correction to improve training dynamics. The method is evaluated on small-scale LLMs.

**Strengths:**

1. The core idea is intuitive. The method leverages previous insights that many channels in LLM layers are redundant and uses that to only compute a subset of channels using FOSL.

2. The method consistently outperforms baseline methods with different model sizes and datasets (Table 2), showing the efficacy of the method.

3. The paper is well-written and easy to follow. The background and related work are well explained, making it easy for readers to understand the proposed method.

4. Ablations are provided to explain how the hyper-parameters (mixing coefficient) were selected.

**Weaknesses:**

1. Results provided are limited. A more extensive evaluation on the zero-shot downstream dataset should be conducted to show that the trained model can be used for downstream tasks.

2. The experiment in section 4.2 is not convincing. The results are mixed, and FOSL has more parameters than LoRA (with the same rank), making direct comparison difficult.

3. In Table 3, the ablation on the mixing coefficient shows that learning the mixing coefficient gives similar performance as using a static coefficient. Do you have any insights?

**Questions:**

1. For predicting remaining channels, have you compared using/learning linear combinations of the computed channels? How does that compare with learning the duplication matrix?

---

> ### Author Response · Authors · 2025-11-21
> **Response to Reviewer SUJz (1/2)**
>
> Dear reviewer, we are really grateful for the thorough review, the positive evaluation, and the valuable concerns you raised. We will do our best to address the different points.
> - **On Weakness 1 (limited downstream evaluation).**
>   We agree that downstream evaluation is important. In the revised paper (see Table 6 in the PDF) we added a zero-shot evaluation of the 1B full-rank and FOSL models using `lm-evaluation-harness` on ARC-Challenge, BoolQ, HellaSwag, MMLU, PIQA, and Winogrande. FOSL closely tracks the full model across these tasks: it improves over the dense baseline on ARC-Challenge and BoolQ, is slightly worse on HellaSwag, PIQA, and Winogrande, and essentially matches it on MMLU, indicating that our perplexity gains translate into comparable zero-shot downstream performance.
>   | Model        | ARC-C (acc / acc\_norm) | BoolQ (acc / acc\_norm) | HellaSwag (acc / acc\_norm) | MMLU (acc / acc\_norm) | PIQA (acc / acc\_norm) | Winogrande (acc / acc\_norm) |
>   |-------------|-------------------------|--------------------------|-----------------------------|------------------------|-------------------------|------------------------------|
>   | 1B Full-rank | 0.180 / 0.300           | 0.300 / --               | 0.390 / 0.440               | 0.232 / --             | 0.700 / 0.700           | 0.550 / --                   |
>   | 1B FOSL     | 0.219 / 0.258           | 0.378 / --               | 0.326 / 0.394               | 0.229 / --             | 0.661 / 0.646           | 0.510 / --                   |
> - **On Weakness 2 (fairness of parameter budgets).**
>   We revised Table 2 so that FOSL uses the same trainable-parameter budgets as the LOST/SLTrain baselines at each LLaMA scale (43M, 94M, 185M, 609M parameters), by adjusting the folding ratio $\rho$ and adapter rank $r$ accordingly. In addition, we added an equal-parameter-budget ablation (new Table 3 in the main text) that sweeps $\rho$ and $r$ while keeping the total parameter count fixed, including a foldable-only variant without the low-rank path. The table below summarizes the 1B-scale ablation at a fixed 609M-parameter budget; similar patterns hold at smaller scales. FOSL’s best configurations (moderate folding $\rho=0.8$–$0.9$ with corresponding ranks) achieve lower perplexity than the more aggressive $\rho=0.99$ setting while staying within the same parameter budget, showing that our gains are not due to extra parameters but due to accurately modeling channel redundancies: by explicitly computing a basis of unique features and replicating them to reconstruct symmetric output channels.
>   | Method                | Folding ratio $\rho$ | Rank $r$ | Params (M) | 1B valid PPL |
>   |-----------------------|----------------------|----------|------------|--------------|
>   | FOSL                  | 0.99                 | 499      | 609        | 14.91        |
>   | FOSL                  | 0.90                 | 382      | 609        | 14.88        |
>   | FOSL                  | 0.80                 | 253      | 609        | 14.99        |
>   | FOSL                  | 0.70                 | 123      | 609        | 15.20        |
>   | FOSL (foldable-only)  | 0.61                 | 0        | 609        | 16.46        |

---

> ### Author Response · Authors · 2025-11-21
> **Response to Reviewer SUJz (2/2)**
>
> - **On Weakness 3 (learnable vs. fixed mixing coefficient).**
>   The differences between fixed $\gamma{=}0.7$ and a trainable per-layer $\gamma$ are indeed modest in Table 3 (e.g., 31.61 vs. 31.45 PPL at 60M, 23.95 vs. 23.82 at 130M, 18.52 vs. 18.23 at 350M), but they are consistent across all scales we evaluated. Intuitively, the mixing coefficient only controls how much of the signal flows through the low-rank vs.\ folded path. Since both branches are expressive and are followed by LayerNorm and other rescalings, the optimizer can compensate for a suboptimal fixed $\gamma$ by adjusting the branch parameters themselves, which reduces the marginal benefit of making $\gamma$ trainable. We keep the per-layer trainable $\gamma$ as our default for slightly better average performance, but emphasize in the revision that FOSL is robust to this choice and that fixed $\gamma{=}0.7$ already performs well.
> - **On the question about linear combinations vs. duplication.**
>   It's an interesting idea and also on our todo list. We have not yet implemented a learned linear combination of computed channels. For LLM MLP layers, such a design would require an additional dense mixing matrix over channels (e.g., of size comparable to the output width), introducing a large number of extra parameters and non-trivial compute on top of the folded branch. In contrast, our duplication operator is parameter-free and cheap, which is important for keeping the parameter and memory budgets competitive in the pre-training regime. We view richer mixing (e.g., learned linear combinations) as a promising but orthogonal direction, and plan to explore it in a close future building on top of the lightweight duplication scheme studied in this paper.
>
> In addition, in the new Appendix part titled training dynamics now reports training dynamics for both the 1B and 130M models, plotting evaluation loss vs. training step for full-rank and FOSL under matched settings; the curves largely overlap, confirming that FOSL preserves stable convergence behavior while offering better or comparable final perplexity under reduced parameter/memory budgets.

---

> > ### Comment · Reviewer_SUJz · 2025-11-24
> > **reply to authors**
> >
> > Thank you for providing additional results and an explanation. Other reviewers have raised valid concerns and limitations about the method. I will keep my original score and will follow the discussion with the other reviewers.

---

### Meta-Review · Area_Chair_5F2f · 2025-12-18

**Summary:**

This paper proposes FOSL, a foldable sparse + low-rank reparameterization for Transformer projections, combining a low-rank adapter with a folded sparse branch that computes a subset of channels and reuses them, together with a variance-preserving rescaling to stabilize channel reuse. The overall reviewer consensus remains negative (with three reject), with the primary concern being the claimed efficiency is not validated in large models. Particularly, the throughput and memory consumption are worse than full-rank baselines, thus questioning the benefits of the proposed method.

**Reviewer Concerns:**

The main concern that actual throughput and memory consumption are higher than full-rank baselines still remains. Although authors claimed that this is an engineering problem and aim to optimise the implementation. No further results are presented.

**Reviewer Scores:**

Reviewer SUJz is the only positive reviewer leaning towards acceptance, yet he remains borderline by considering the concerns of other reviewers. Reviewer 1eKn may have increased the rating after the rebuttal. However, both Reviewer NvE8 and gyFj are mainly concerned about the actual model latency. The rebuttal did not fully address the concern.

---

### Decision · Program_Chairs · 2026-01-26

Reject